# ChartE³: A Comprehensive Benchmark for End-to-End Chart Editing

Shuo Li [* 1]  Jiajun Sun [* 1]  Zhekai Wang [* 1]  Xiaoran Fan [1]  Hui Li [1]  Dingwen Yang [1]  Zhiheng Xi [1]
Yijun Wang [2]  Zifei Shan [2]  Tao Gui [1 3]  Qi Zhang [1]  Xuanjing Huang [1]

## Abstract

Charts are a fundamental visualization format for structured data analysis. Enabling end-to-end chart editing according to user intent is of great practical value, yet remains challenging due to the need for both fine-grained control and global structural consistency. Most existing approaches adopt pipeline-based designs, where natural language or code serves as an intermediate representation, limiting their ability to faithfully execute complex edits. We introduce ChartE³, an **E**nd-to-**E**nd Chart **E**diting benchmark that directly evaluates models without relying on intermediate natural language programs or code-level supervision. ChartE³ focuses on two complementary editing dimensions: local editing, which involves fine-grained appearance changes such as font or color adjustments, and global editing, which requires holistic, data-centric transformations including data filtering and trend line addition. ChartE³ contains over 1,200 high-quality samples constructed via a well-designed data pipeline with human curation. Each sample is provided as a triplet of a chart image, its underlying code, and a multimodal editing instruction, enabling evaluation from both objective and subjective perspectives. Extensive benchmarking of state-of-the-art multimodal large language models reveals substantial performance gaps, particularly on global editing tasks, highlighting critical limitations in current end-to-end chart editing capabilities.[1]

## 1. Introduction

Charts are one of the most widely used visualization formats for presenting structured data in scientific research, business analysis, and decision-making workflows. Unlike natural images, charts encode precise numerical values, semantic structures, and visual conventions, making their correct interpretation and manipulation essential for downstream data analysis tasks. As multimodal large language models (MLLMs) (OpenAI, 2024) continue to advance in visual understanding and reasoning, chart understanding has emerged as a critical testbed for evaluating their ability to jointly reason over visual and structured information.

Recent progress in chart-related research can be broadly categorized into two main lines of work. The first line focuses on chart understanding, including tasks such as chart classification, data extraction, question answering, and numerical reasoning over plotted values. These studies (Masry et al., 2022; Xu et al., 2024; Xia et al., 2025) aim to recover the semantics and underlying data encoded in chart images, and have led to substantial improvements in models' abilities to interpret structured visual information. The second line of work addresses chart editing through code-mediated pipelines. In this paradigm, models first translate user intent into chart-specific code or program representations—such as plotting scripts or configuration files—which are then executed by a rendering engine to produce the edited chart image (Zhao et al., 2025; Yang et al., 2025b; Wu et al., 2024; Chen et al., 2025; Goswami et al., 2025). This formulation naturally leverages the structured nature of charts and has shown promising results in constrained editing scenarios. However, such approaches primarily evaluate a model's ability to generate syntactically and semantically correct code, rather than its capacity to perform true end-to-end chart editing, where an input chart is directly transformed into an edited output. Moreover, the reliance on intermediate code representations decouples visual perception from editing execution, making it difficult to assess how well models jointly reason over visual appearance, data semantics, and editing intent. As a result, the end-to-end editing capability of multimodal models remains largely unexplored and insufficiently evaluated as shown in Table 1.

To address these limitations, we introduce ChartE³, an end-

---

[*]Equal contribution  [1]Fudan University  [2]Tencent  [3]Shanghai Innovation Institute. Correspondence to: <lis23@m.fudan.edu.cn>, <tgui@fudan.edu.cn>.

*Proceedings of the 43rd International Conference on Machine Learning*, Seoul, South Korea. PMLR 306, 2026. Copyright 2026 by the author(s).

[1]This Benchmark is open sourced at https://github.com/galactic123/ChartE3

*Table 1.* Comparison of our proposed benchmark with existing chart generation benchmarks. While prior benchmarks mainly target captioning, QA, or chart-to-code generation, they offer limited support for end-to-end chart editing. FigEdit is a concurrent benchmark, in contrast, **Chart E**[3] provides broader coverage of charts and places stronger emphasis on global editing tasks with reference target images, enabling more reliable evaluation of complex chart edits.

| Name | Output Format | Image Source | Chart Types | w/ Ref. Image | Local Editing | Global Editing | | | |
|---|---|---|---|---|---|---|---|---|---|
| | | | | | | Conversion | Order | Filtering | Reference |
| ChartCraft (Yan et al., 2024) | Json | Syn. | 5 | ✓ | ✓ | ✗ | ✗ | ✓ | ✗ |
| Plot2Code (Wu et al., 2024) | Code | Syn. | 6 | ✓ | ✗ | ✗ | ✗ | ✗ | ✗ |
| ChartX (Xia et al., 2025) | Code | Syn. | 18 | ✓ | ✓ | ✗ | ✗ | ✗ | ✗ |
| AcademiaChart (Zhang et al., 2024) | Code | Real | 6 | ✓ | ✗ | ✗ | ✗ | ✗ | ✗ |
| ChartM[3] (Yang et al., 2025b) | Code | Syn. | 10 | ✓ | ✓ | ✓ | ✗ | ✗ | ✗ |
| ChartEdit (Zhao et al., 2025) | Code | Real | 19 | ✓ | ✓ | ✓ | ✗ | ✗ | ✗ |
| FigEdit* (Li et al., 2025) | Figure | Syn. | 7/10 | ✗ | ✓ | ✗ | ✗ | ✗ | ✗ |
| **Chart E**[3] **(Ours.)** | **Figure** | **Syn.&Real** | **10/42** | ✓ | ✓ | ✓ | ✓ | ✓ | ✓ |

to-end Chart Editing benchmark designed to systematically evaluate chart editing capabilities of multimodal models under a true image-to-image setting. Unlike prior benchmarks that operate through intermediate code representations (e.g., ChartM3 (Yang et al., 2025b), ChartX (Xia et al., 2025)), where editing instructions explicitly reference code variables or rendering patterns that are inaccessible to image editing models, ChartE[3] focuses on fully image-based editing. It evaluates whether models can directly manipulate visual charts while preserving perceptual fidelity, semantic correctness, and structural consistency. ChartE[3] is constructed through a carefully designed five-stage data pipeline to ensure both diversity and annotation quality. Specifically, we first collect raw chart data from diverse sources and chart types. We then perform image diversity filtering to reduce redundancy and ensure broad coverage of visual layouts and styles. Based on the filtered charts, we obtain paired chart–code representations, which serve as a reliable reference for subsequent edit generation. Using these paired representations, we generate edited chart data that reflect a wide range of realistic editing intents. Finally, we apply automatic filtering followed by human verification to remove invalid or ambiguous samples and to ensure the correctness and consistency of annotations. The benchmark is organized around two major categories of editing tasks: local editing and global editing. Local editing focuses on fine-grained, appearance-level modifications, such as adjusting fonts, colors, or annotations, while global editing requires holistic, data-centric transformations, including data filtering, aggregation, and the addition of derived visual elements. These two categories are further divided into 12 fine-grained task types, capturing a broad spectrum of chart editing tasks. In total, ChartE[3] contains over 800 unique chart images and more than 1,200 annotated editing samples. We employ objective metrics, including SSIM and CLIP-based similarity, to measure visual consistency, and complement them with GPT-based subjective scoring to assess semantic correctness and edit faithfulness.

Using this benchmark, we conduct extensive evaluations

of both open-source and closed-source multimodal models. Our results reveal several notable findings: closed-source models often outperform open-source counterparts on end-to-end chart editing; local editing tasks are consistently easier than global, data-level edits; and error analysis shows that many failures stem from limitations in text generation and instruction grounding, even when visual understanding is correct.

In summary, our main contributions are:

- We introduce a new paradigm for end-to-end chart editing, emphasizing direct transformation without intermediate code or program representations;

- We present ChartE[3], a comprehensive benchmark for evaluating end-to-end chart editing across diverse local and global editing tasks;

- We perform extensive benchmarking and analysis of current multimodal models, uncovering systematic strengths, weaknesses, and failure modes that point to future research directions in end-to-end chart editing.

## 2. Related Work

**Multimodal Large Language Models** Multimodal large language models (MLLMs) have recently attracted significant attention as a unified framework for visual–language understanding, with representative systems such as GPT-4o (OpenAI, 2024) demonstrating strong generalization across heterogeneous tasks. These models typically combine pretrained vision backbones with large-scale language models, enabling the alignment and interaction of visual signals and linguistic representations within a single reasoning pipeline. Early studies, most notably CLIP (Radford et al., 2021), pioneered large-scale contrastive vision–language pretraining, providing a fundamental mechanism for cross-modal representation alignment. LLaVA (Liu et al., 2024c;b) further simplifies multimodal integration through a lightweight projection interface and

staged training strategy, achieving strong performance with improved spatial reasoning. More recent approaches, such as MouSi (Fan et al., 2024) and Cambrian-1 (Tong et al., 2024), explore heterogeneous or vision-centric encoder designs to enhance multimodal perception. In parallel, a series of open and proprietary systems, including Qwen-VL (Bai et al., 2023; Wang et al., 2024a; Bai et al., 2025), InternLM-XComposer (Zhang et al., 2023; Dong et al., 2024), and InternVL (Chen et al., 2023; 2024), largely follow the LLaVA-style architecture while steadily pushing the performance frontier on standard multimodal benchmarks. In addition to models primarily designed for visual understanding and reasoning, a substantial body of work augments MLLMs with image generation capabilities, enabling unified modeling of visual understanding and generation; representative examples include BAGEL (Deng et al., 2025) and Omni-Gen2 (Wu et al., 2025b).

**Image Editing** Image editing has been extensively studied, with early methods focusing on pixel-level manipulation and rule-based pipelines. With the advent of deep generative models, learning-based approaches have enabled more flexible and semantic editing, including GAN-based image manipulation (Isola et al., 2017; Zhu et al., 2017) and diffusion-based editing frameworks (Rombach et al., 2022; Hertz et al., 2023). More recently, large language models have been incorporated to support instruction-driven image editing, where edits are specified via natural language prompts and executed through generative models (Brooks et al., 2023; Pan et al., 2023; Yu et al., 2025). However, most existing methods either rely on ambiguous textual instructions or operate in a generation-centric paradigm, without explicitly evaluating whether models can perform end-to-end, fine-grained, and faithful edits on structured visual content such as charts.

**Chart Understanding and Editing** Chart understanding has been extensively studied through a series of benchmarks that progressively increase task complexity. ChartQA (Masry et al., 2022) and MMC(Liu et al., 2024a) introduced large-scale visual question answering over charts, requiring joint visual perception and logical reasoning, while subsequent datasets further exposed the gap between multimodal models and human performance on scientific and expert-level charts (Wang et al., 2024b). Beyond understanding, chart generation tasks such as text-to-chart and chart-to-code focus on automatic chart synthesis from textual or structured specifications (Tang et al., 2025; Yang et al., 2025a). More recently, chart editing has been explored in interactive settings, where models modify existing charts by revising intermediate rendering code according to natural language instructions (Zhao et al., 2025). Although these code-mediated pipelines ensure executability and structural validity, they often struggle with instruction

ambiguity and provide limited evaluation of perceptual quality. Given that charts encode structured data, calibrated axes, and embedded text, editing charts poses challenges distinct from natural image editing (Brooks et al., 2023). In contrast to prior work, we focus on benchmarking end-to-end chart image editing, emphasizing direct visual manipulation without relying on intermediate code representations.

## 3. ChartE$^3$ Benchmark

### 3.1. Task Definition

Let a chart be represented as an image $I \in \mathcal{I}$, generated from a chart specification or rendering code $C \in \mathcal{C}$ through a deterministic rendering function

$$I = \mathcal{R}(C), \tag{1}$$

where $\mathcal{R}(\cdot)$ denotes a chart rendering engine (e.g., matplotlib (Hunter, 2007) or Vega-Lite (Satyanarayan et al., 2017)).

**Code-mediated Chart Editing.** Most existing chart editing methods formulate editing as a code transformation problem. Given an original chart image $I$, its corresponding code $C$, and a user instruction $u \in \mathcal{U}$, the model predicts an edited code

$$C' = f_{\text{code}}(I, C, u), \tag{2}$$

which is subsequently rendered into the edited chart image

$$I' = \mathcal{R}(C'). \tag{3}$$

This paradigm emphasizes executability and syntactic correctness in code space, while the visual outcome is indirectly constrained by the rendering engine. As a result, perceptual faithfulness and fine-grained visual consistency are not explicitly optimized.

**End-to-End Chart Image Editing.** In contrast, we define chart editing as a direct visual transformation task. Given an original chart image $I$ and an editing instruction $u$, the model directly generates the edited image

$$I' = f_{\text{e2e}}(I, u), \tag{4}$$

without relying on any intermediate code representation. This end-to-end formulation requires the model to jointly understand structured chart elements, such as data marks, axes, legends, and textual annotations, and to execute edits directly in image space. Under this formulation, an end-to-end chart editing benchmark evaluates whether the predicted image $I'$ is visually faithful and semantically aligned with the intended edit, without assuming access to or supervision from chart-rendering code. This setting complements existing code-centric benchmarks by explicitly targeting perceptual quality and holistic chart usability.

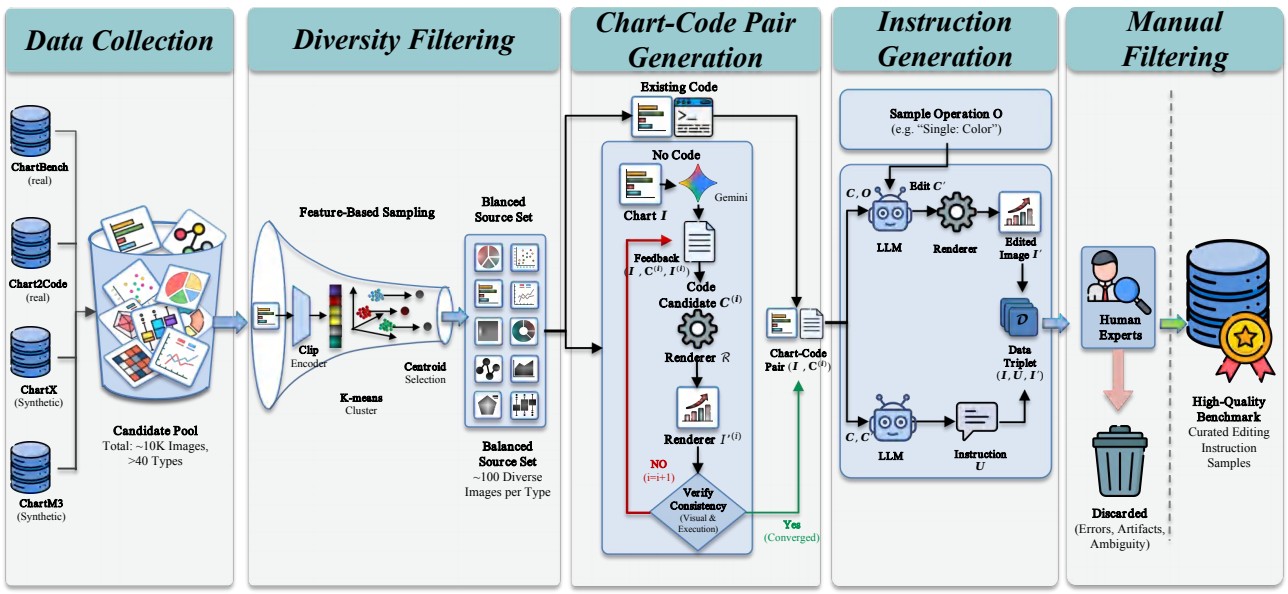

*Figure 1.* Overview of the five-stage data construction pipeline for ChartE[3].

## 3.2. Data Pipeline

To construct a high-quality benchmark for end-to-end chart editing, we design a five-stage data construction pipeline that systematically collects, processes, and verifies chart editing samples, ensuring that the resulting dataset supports a robust and reliable evaluation of chart editing models.

**Data Collection.** To ensure broad coverage and realism, we collect chart images from multiple existing chart-related datasets, including both real-world and synthetic sources. Specifically, we incorporate charts from ChartBench (Xu et al., 2024) and Chart2Code (Tang et al., 2026) as real-world data, which contain diverse chart styles collected from practical applications. In addition, we leverage synthetic chart datasets, including ChartX (Xia et al., 2025) and ChartM[3] (Yang et al., 2025b), to supplement coverage of underrepresented chart types and visual configurations. From these sources, we curate a unified pool of chart im-

*Table 2.* Collect the data from the original chart dataset.

| Name | Source | Select Num. | Chart Types |
|------|--------|-------------|-------------|
| ChartBench | Real | 2.1K | 9/42 |
| ChartX | Syn. | 6.0k | 18 |
| ChartM[3] | Syn. | 1.0k | 10 |
| Chart2code | Real | 0.9k | 22 |

ages spanning a wide range of chart types, visual styles, and data distributions. As summarized in Table 2, our collection includes 10K candidate images across more than 40 chart types.

**Diversity Filtering** Following the chart type defined in Section A, we first group all candidate chart images into 10 chart types.

To promote visual diversity and reduce redundancy within each type, we perform feature-based sampling. Specifically, given a set of chart images $\{I_i\}_{i=1}^N$ in a type, we extract visual embeddings using a CLIP (Radford et al., 2021) image encoder:

$$\mathbf{z}_i = \text{CLIP}_{\text{img}}(I_i), \tag{5}$$

where $\mathbf{z}_i \in \mathbb{R}^d$ denotes the image feature representation.

We then apply $k$-means clustering in the embedding space:

$$\{\mathcal{C}_j\}_{j=1}^k = k\text{-means}(\{\mathbf{z}_i\}_{i=1}^N), \tag{6}$$

and select representative images by choosing the cluster centroids:

$$I_j^* = \arg\min_{I_i \in \mathcal{C}_j} \|\mathbf{z}_i - \boldsymbol{\mu}_j\|_2, \tag{7}$$

where $\boldsymbol{\mu}_j$ denotes the centroid of cluster $\mathcal{C}_j$.

After diversity-aware sampling, each chart type contains approximately 100 representative images, forming a balanced and visually diverse source dataset for subsequent chart editing annotation.

**Chart-Code Pair Generation.** To support structured editing instruction generation, we associate each chart image with a corresponding chart-rendering code representation. For datasets that already provide paired chart images and code, we directly adopt the original chart-code pairs.

For chart images without accompanying code, we employ Gemini-2.5-Pro (DeepMind, 2025) to synthesize chart-

rendering code, distinguishing it from the GPT used in the evaluation to prevent data leakage. Given a chart image $I$, the model predicts an initial code candidate

$$C^{(0)} = f_{\text{LLM}}(I), \tag{8}$$

which is expected to reproduce the input chart when rendered.

To improve code fidelity, we adopt a reflection-style generation process. Specifically, the generated code is rendered to obtain $\hat{I}^{(t)} = \mathcal{R}(C^{(t)})$ and verified through two criteria: (1) whether the code successfully executes, and (2) whether the rendered image is visually consistent with the original chart (judged by whether CLIP similarity higher than 0.7). If either condition is violated, the model is prompted to revise the code based on the detected discrepancy:

$$C^{(t+1)} = f_{\text{LLM}}\big(I, C^{(t)}, \hat{I}^{(t)}\big), \tag{9}$$

where the iteration continues until convergence or a maximum number of refinement steps ($t \leq 3$) is reached. In the end, most (over 99%) of the examples are retained.

The final code $C^*$ satisfies

$$\mathcal{R}(C^*) \approx I, \tag{10}$$

and is used solely as an intermediate structural representation to facilitate the generation of precise editing instructions in subsequent stages. Importantly, this code serves only as an annotation aid.

**Instruction Generation.** Based on the chart–code pairs obtained in the previous stage, we generate editing instructions that specify the intended chart modifications. Given an original chart image $I$ and its associated code representation $C$, we first sample an editing operation $o$ from a predefined set of chart editing primitives, such as modifying visual attributes, updating data selections, or introducing additional chart components. We organize editing instructions into two categories: **local** and **global** editing, based on the scope of changes they induce. Local editing targets specific visual elements while preserving the overall chart structure, such as modifying numeric values, colors, titles, fonts, legends, axis labels, or low-level visual styles. In contrast, global editing introduces structural or semantic transformations that affect the entire chart, including chart type conversion, data reordering, data filtering, and the addition of reference elements (e.g., trend or average lines).

Conditioned on the selected task, gemini generates an edited code $C'$:

$$C' = f_{\text{LLM}}(C, o), \tag{11}$$

which is then rendered into the edited chart image:

$$I' = \mathcal{R}(C'). \tag{12}$$

To obtain the corresponding editing instruction, we ask the model to verbalize the transformation from $C$ to $C'$ as a natural language instruction:

$$u = g_{\text{LLM}}(C, C'). \tag{13}$$

The resulting instruction $u$ describes the intended edit in a concise and executable manner, while remaining agnostic to the underlying code syntax.

Each data sample is thus represented as a triplet $(I, u, I')$, where $u$ represents the editing instruction, and $I'$ serves as the reference edited image.

**Manual Filtering** To ensure annotation quality and benchmark reliability, we perform a final manual filtering stage in which three human experts inspect each generated editing sample. Samples are discarded if they exhibit incorrect or incomplete edits, severe visual artifacts, or significant occlusions that hinder chart readability, as well as low-quality images caused by rendering failures or ambiguous editing outcomes that cannot be unambiguously interpreted from the instruction. For a small subset of recoverable cases (e.g., minor visual glitches or formatting issues), human experts manually correct the outputs to match the intended editing instruction, while preserving the original data semantics.

Disagreements among annotators are resolved through discussion to reach a consensus. This manual verification step further refines the dataset and ensures that the retained samples accurately reflect well-defined chart editing intentions with clear images.

### 3.3. Data Statistic

Finally, our Benchmark contains over 800 images and over 1200 instructions. The detailed statistics are shown in Table 4. And the distribution of chart types and tasks in our data is shown in the Section B.

### 3.4. Evaluation Metrics

To comprehensively evaluate end-to-end chart image editing, we adopt a hybrid evaluation protocol that combines objective visual similarity metrics with subjective semantic assessment. Given the original chart image $I$, the model output $\hat{I}$, and the reference edited image $I'$, we evaluate model performance from both perceptual and semantic perspectives.

**Objective Metrics.** We employ five widely-used objective metrics to measure visual similarity between $\hat{I}$ and $I'$:

- **SSIM** (Wang et al., 2004), which measures structural similarity in terms of luminance, contrast, and structure.

*Table 3.* Definition and examples of chart editing tasks.

| Task | Definition | Example |
|---|---|---|
| Single: Numeric | Change the value or proportion of a specific data point. | Increase the second bar value by 30%. |
| Single: Scale | Modify data range, interval, or tick label formatting. | Set Y-axis range to [0, 300]. |
| Single: Color | Modify the color or color palette of specific elements. | Change the line color to vibrant green. |
| Single: Title | Modify, add, or delete the main or sub-title. | Change title to "Sales 2024". |
| Single: Font | Modify the font size of any text element. | Set all axis label fonts to 12pt. |
| Single: Legend | Change the legend position, title, or labels. | Move legend to bottom right. |
| Single: Axis | Modify the names of the X-axis or Y-axis labels. | Rename X-axis to "Months". |
| Single: Visual Elements | Modify visual style such as stroke thickness, markers, or borders. | Double the line thickness and use diamond markers. |
| Global: Conversion | Convert the current chart type to another type. | Convert this bar chart into a line chart. |
| Global: Order | Reorder data points or subplots. | Sort bars in descending order. |
| Global: Filtering | Filter data based on specific conditions. | Exclude data points below the average. |
| Global: Reference Elements | Add or modify reference elements (e.g. average lines) | Add a red dashed horizontal line at the mean value. |

*Table 4.* The statistics of Chart E$^3$ benchmark.

| Statistic | Number |
|---|---|
| Total Images | 808 |
| Total Instructions | 1211 |
| Chart Types | 10/42 |
| Editing Tasks | 12 |
| Average Instruction Length | 22.03 |
| Average Image Pixel | 1584*1161 |
| Local Editing Instruction Length | 19.27 |
| Global Editing Instruction Length | 26.41 |
| Local Editing Images | 565 |
| Global Editing Images | 423 |

- **PSNR** (Horé & Ziou, 2010), which quantifies pixel-level reconstruction fidelity.

- **CLIP Similarity** (Radford et al., 2021), computed as the cosine similarity between CLIP image embeddings of $\hat{I}$ and $I'$, capturing high-level semantic alignment.

- **DINO Similarity** (Siméoni et al., 2025), which evaluates perceptual similarity using self-supervised visual representations.

- **LPIPS** (Zhang et al., 2018), which measures perceptual distance aligned with human judgments.

These metrics jointly assess low-level visual fidelity and high-level semantic consistency of the edited charts. The definitions and implementations of these metrics can be found in the Section E.

**Subjective Metrics.** Objective similarity alone cannot fully capture whether an edit is semantically correct or faithful to the user instruction. Therefore, we further adopt two subjective evaluation criteria using a GPT-based judge. The GPT-5.1 is provided with the original image $I$, the generated image $\hat{I}$, and the reference image $I'$, and assigns scores along the following dimensions:

- **Correctness**: whether the chart elements specified by the editing instruction are edited correctly and completely.

- **Consistency**: whether chart elements not involved in the instruction remain visually consistent with the original image.

For objective evaluation, all images are resized to a unified resolution, defined as the reference image size or a prescribed input size (e.g., 336px for CLIP-based metrics). In contrast, for subjective evaluation, images are kept at their original resolutions to preserve perceptual details. Together, these subjective scores assess both edit faithfulness and unintended changes, complementing the objective metrics for a more reliable evaluation of end-to-end chart image editing performance.

## 4. Experiments

### 4.1. Setup

We evaluate a diverse set of state-of-the-art image editing models, including both closed-source and open-source models. Specifically, we consider two strong closed-source models, **GPT-Image-1.5** (OpenAI, 2025) and **Nano Banana** (Google DeepMind, 2025), as well as five representative open-source models: **Step1X-Edit** (Liu et al., 2025), **BAGEL** (Deng et al., 2025), **OmniGen2** (Wu et al., 2025b), **Qwen-Image-Edit** (Wu et al., 2025a), and **Instruct-Pix2Pix** (Brooks et al., 2023).

All models are evaluated on the proposed benchmark under a unified end-to-end chart image editing setting. For each model, we use its officially released inference code and recommended configurations to avoid implementation bias.

Experiments are conducted on a server equipped with $8\times$ NVIDIA A100 GPUs. For models that support multi-GPU inference, we enable parallel execution following their official implementations. All models are evaluated using the same input images and editing instructions, and their outputs are assessed using the evaluation metrics described in

Section 3.4.

## 4.2. Main Results

Table 5 summarizes the evaluation results of different chart image editing models on our benchmark, including five objective metrics and two subjective metrics.

Overall, closed-source models exhibit strong and stable performance across both objective and subjective evaluations. Nano Banana achieves the best overall results, obtaining the highest PSNR and DINO similarity, the lowest LPIPS, and the highest scores on both Correctness and Consistency. GPT-Image-1.5 also performs competitively, particularly on semantic and perceptual similarity metrics such as CLIP and DINO, indicating strong alignment between edited outputs and reference images.

Among open-source models, Qwen-Image-Edit and BAGEL demonstrate relatively balanced performance. Qwen-Image-Edit achieves comparable SSIM and PSNR to closed-source models, while BAGEL maintains reasonable perceptual quality with moderate LPIPS scores. However, both models fall behind closed-source approaches in subjective evaluations, suggesting difficulties in accurately following editing instructions and preserving non-edited chart elements.

Other open-source baselines, including InstructPix2Pix, OmniGen2, and Step1X-Edit, show noticeably weaker performance across most metrics, especially in perceptual similarity and subjective scores. This indicates that general-purpose image editing models struggle to handle the fine-grained, structure-sensitive requirements of chart image editing.

In summary, the results reveal a clear performance gap between closed-source and open-source models on end-to-end chart image editing, particularly in terms of instruction correctness and visual consistency.

In addition, we observe a strong consistency between objective and subjective evaluations. Models achieving higher scores on objective metrics (e.g., SSIM, CLIP, DINO, and LPIPS) also tend to obtain better Correctness and Consistency scores in GPT-aligned judgment. This alignment indicates that improvements in low-level visual fidelity and high-level semantic similarity are generally correlated with more faithful execution of editing instructions and better preservation of non-edited chart elements.

## 4.3. Analysis

**Performance across Editing Tasks**    Table 6 reveals clear performance disparities across different editing tasks. Appearance-centric edits achieve the highest scores overall: for example, Nano Banana reaches correctness/consistency scores of 4.82/4.82 on Color and 4.74/4.95 on Axis, while GPT-Image-1.5 attains 4.79/4.89 on Title. In contrast, global editing remains significantly more challenging. Even the strongest model, Nano Banana, drops to 3.42/4.42 on Order and 3.59/4.07 on Filtering, with GPT-Image-1.5 further declining to 2.44/3.35 and 3.05/3.42, respectively. Open-source models exhibit substantially lower scores across these types; for instance, most achieve correctness below 2.0 on ordering and filtering tasks, indicating frequent failures in executing data-dependent edits. Notably, reference element editing is relatively more manageable, where Nano Banana achieves 4.69/4.90, suggesting that additive global edits are easier than local ones. Overall, while model rankings remain consistent across types, the pronounced gap between local edits and structurally global edits highlights a fundamental limitation of current end-to-end image editing models in handling chart-level data semantics.

**Performance across Chart Types**    Table 7 reports model performance across different chart types. Overall, closed-source models substantially outperform open-source ones across all chart types, with Nano Banana consistently achieving the highest scores. In particular, Nano Banana performs strongly on common chart types such as bar, heatmap, and scatter plots, reaching correctness/consistency scores of 4.38/4.71, 4.49/4.68, and 4.43/4.79, respectively. GPT-Image-1.5 shows competitive and stable performance across most chart types, with especially strong results on node-based charts (4.24/4.57) and line charts (4.00/4.34).

Across all models, simpler and more standardized chart types tend to yield higher performance. Bar, line, and scatter charts consistently receive higher scores than more specialized or less frequently used types, such as radar and box plots. For example, the average correctness score of open-source models on radar charts remains below 2.0, while their performance on bar charts is relatively higher but still limited. Heatmaps also exhibit a clear performance gap between closed- and open-source models, suggesting that dense visual encodings and color–value mappings pose additional challenges.

Notably, performance on the Other chart type remains comparatively stable for closed-source models (e.g., 4.33/4.64 for Nano Banana), indicating stronger generalization to diverse or less common chart formats. In contrast, open-source models struggle to maintain consistency across chart types, often failing to preserve structural coherence when editing complex or uncommon layouts. These results suggest that robustness across chart types remains a key bottleneck for current end-to-end chart editing models.

**Alignment between GPT Score and Human**    To assess the reliability of GPT-based subjective evaluation, we analyze its alignment with human judgments on a subset of 140 randomly selected editing samples. For each sample,

*Table 5.* Overview the evaluation results of the model. Cor. and Con. represent the correctness and consistency score evaluated by GPT.

| Model | Objective | | | | | Subjective | |
|---|---|---|---|---|---|---|---|
| | SSIM | PSNR | CLIP | DINO | LPIPS↓ | Cor. | Con. |
| GPT-Image-1.5 | 0.79 | 15.05 | **0.96** | 0.95 | 0.36 | 3.79 | 4.28 |
| Nano Banana | **0.83** | **17.53** | 0.96 | **0.96** | **0.23** | **4.18** | **4.65** |
| BAGEL | 0.82 | 16.28 | 0.89 | 0.93 | 0.28 | 1.84 | 3.23 |
| InstructPix2Pix | 0.78 | 14.35 | 0.66 | 0.88 | 0.38 | 1.49 | 2.08 |
| OmniGen2 | 0.55 | 8.70 | 0.78 | 0.82 | 0.52 | 1.48 | 2.18 |
| Step1X-Edit | 0.77 | 14.35 | 0.86 | 0.89 | 0.38 | 1.72 | 2.46 |
| Qwen-Image | 0.83 | 16.83 | 0.93 | 0.93 | 0.28 | 2.58 | 3.15 |

*Table 6.* The evaluation results of the model on different editing tasks.

| Model | Local | | | | | | | | Global | | | |
|---|---|---|---|---|---|---|---|---|---|---|---|---|
| | Numeric | Scale | Color | Title | Font | Legend | Axis | Visual | Conversion | Order | Filtering | Reference |
| GPT-Image-1.5 | 2.94/4.15 | 3.37/4.12 | 4.28/4.55 | **4.79/4.89** | 4.08/4.53 | 4.00/4.57 | 4.10/4.55 | 4.22/4.61 | 3.73/3.82 | 2.44/3.35 | 3.05/3.42 | 4.28/4.64 |
| Nano Banana | **3.56/4.61** | **3.80/4.56** | **4.82/4.82** | 4.52/4.87 | **4.11/4.74** | **4.01/4.66** | **4.74/4.95** | **4.47/4.69** | **4.02/4.19** | **3.42/4.42** | **3.59/4.07** | **4.69/4.90** |
| BAGEL | 1.46/4.05 | 1.47/3.73 | 2.90/3.56 | 1.98/3.29 | 2.74/3.93 | 1.76/2.77 | 1.25/3.35 | 2.41/3.22 | 1.70/1.67 | 1.15/3.09 | 1.05/3.12 | 1.86/3.06 |
| InstructPix2Pix | 1.60/2.05 | 1.84/2.19 | 2.21/2.07 | 1.29/1.79 | 1.65/1.94 | 1.44/2.08 | 1.25/2.30 | 1.68/2.02 | 1.01/2.46 | 1.41/2.49 | 1.17/1.95 | 1.23/1.77 |
| OmniGen2 | 1.29/2.39 | 1.32/2.26 | 2.08/2.43 | 1.99/2.58 | 2.17/2.63 | 1.36/1.95 | 1.18/2.04 | 1.49/1.95 | 1.20/2.05 | 1.14/2.27 | 1.14/1.94 | 1.38/1.84 |
| Step1X-Edit | 1.58/2.79 | 1.36/2.28 | 2.53/3.08 | 1.51/2.16 | 2.11/2.85 | 1.64/2.42 | 1.69/3.13 | 1.89/2.29 | 1.40/1.76 | 1.49/2.47 | 1.20/2.00 | 1.78/2.23 |
| Qwen-Image-Edit | 1.67/2.82 | 1.62/2.27 | 3.47/3.91 | 4.02/4.45 | 2.83/3.27 | 2.79/3.33 | 3.26/4.08 | 3.56/3.92 | 1.87/2.05 | 1.29/1.95 | 1.20/2.30 | 2.98/3.40 |

*Table 7.* The evaluation results of the model on different chart types.

| Model | Area | Bar | Box | Heatmap | Line | Node | Pie | Radar | Scatter | Other |
|---|---|---|---|---|---|---|---|---|---|---|
| GPT-Image-1.5 | 3.65/4.15 | 3.72/4.30 | 3.70/4.39 | 3.64/3.94 | 4.00/4.34 | 4.24/4.57 | 3.90/4.38 | 3.36/4.14 | 3.94/4.30 | 3.93/4.36 |
| Nano Banana | **3.82/4.49** | **4.38/4.71** | **3.94/4.70** | **4.49/4.68** | **4.30/4.70** | **4.25/4.54** | **4.09/4.73** | **3.90/4.48** | **4.43/4.79** | **4.33/4.64** |
| BAGEL | 1.71/3.39 | 1.75/3.31 | 1.67/3.30 | 1.86/3.26 | 1.71/3.18 | 2.53/3.28 | 1.91/3.01 | 1.55/2.88 | 1.96/3.33 | 1.94/3.36 |
| InstructPix2Pix | 1.48/2.32 | 1.52/2.23 | 1.33/2.20 | 1.51/2.03 | 1.42/2.10 | 1.61/1.80 | 1.51/1.69 | 1.63/2.02 | 1.29/2.12 | 1.59/2.27 |
| OmniGen2 | 1.47/2.53 | 1.45/2.22 | 1.48/2.32 | 1.31/2.51 | 1.39/1.69 | 1.83/2.42 | 1.55/2.13 | 1.47/1.92 | 1.42/1.63 | 1.54/2.29 |
| Step1X-Edit | 1.65/3.02 | 1.58/2.20 | 1.46/2.09 | 1.72/2.72 | 1.62/2.20 | 2.24/2.64 | 1.92/2.67 | 1.63/2.37 | 1.69/2.24 | 1.77/2.43 |
| Qwen-Image-Edit | 2.54/3.20 | 2.53/3.18 | 2.31/3.04 | 2.40/3.02 | 2.61/3.39 | 3.30/3.71 | 2.49/3.03 | 2.35/2.80 | 2.68/3.21 | 2.80/3.21 |

*Table 8.* Results of the consistency between GPT score and human evaluation.

| Metric | Pairwise Accuracy | NDCG@7 |
|---|---|---|
| Correctness | 0.79 | 0.92 |
| Consistency | 0.87 | 0.97 |

human annotators are asked to rank the model-generated images, and the resulting rankings are compared against those induced by GPT scores. Alignment is evaluated using Pairwise Accuracy (Deutsch et al., 2023) and NDCG@7 (Wang et al., 2013). As shown in Table 8, GPT scores demonstrate strong agreement with human judgments across both evaluation criteria. For Correctness, GPT achieves a pairwise accuracy of 0.79 and an NDCG@7 of 0.92, indicating reliable assessment of whether the intended edits are correctly executed. Agreement is even stronger for Consistency, reaching 0.87 in pairwise accuracy and 0.97 in NDCG@7, suggesting that GPT accurately captures the preservation of non-edited chart elements. These results indicate that GPT-based scoring provides a dependable and scalable proxy for

human evaluation in end-to-end chart editing.

**More Analysis** We provide more analysis in the Sections F to H, including the comparison of different image editing paradigms, error analysis and case study.

## 5. Conclusion

This work introduces ChartE[3], a benchmark for systematically evaluating end-to-end chart editing, where models directly transform input charts into edited outputs without relying on intermediate code representations. By covering diverse chart types, editing tasks, and both local and global editing categories, ChartE[3] enables comprehensive assessment using a hybrid evaluation protocol that combines objective visual metrics with GPT-based subjective scoring, whose alignment with human judgments is empirically validated. Extensive experiments on both closed- and open-source models reveal that while current models perform reasonably well on localized visual edits, they struggle with data- and structure-centric modifications, leading to

frequent visual, data, and structural errors. These findings highlight fundamental limitations of existing multimodal models in structure-aware and semantically consistent chart editing, and position ChartE[3] as a valuable benchmark for advancing robust end-to-end chart editing models.

## Impact Statement

This paper presents work whose goal is to advance the field of Machine Learning. There are many potential societal consequences of our work, none which we feel must be specifically highlighted here.

## Acknowledgement

The authors wish to thank the anonymous reviewers for their helpful comments. This work was partially funded by National Natural Science Foundation of China (No. 62476061, 62576106, 62376061).

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

## A. Chart Types Definitions

*Table 9.* The existing charts are categorized into 10 types of charts.

| Type | Description |
|---|---|
| Area | Normal Area, Percent Area, Stack Area |
| Bar | Normal Bar, Horizontal Bar, Vertical Bar, Multi Bar, Percent Bar, Stacked Bar, 3D Bar, Num Bar, Histogram, Anno Bar, Combination Bar |
| Box | Normal Box, Stock Box, Horizontal Box, Vertical Box |
| Heatmap | Normal Heatmap |
| Line | Normal Line, Multi Line, Anno Line, Num Line, Err Line |
| Node | Normal Node |
| Other | Violin, Bubble, Candelstick, Funnel, Multi Axes, Treemap, Rose |
| Pie | Normal Pie, Ring Pie, Anno Pie, Sector Pie |
| Radar | Normal Radar, Multi Radar, Anno Radar |
| Scatter | Normal Scatter, 3D Scatter, Smooth Scatter |

The chart types Definitions are shown in Table 9.

## B. Detail Chart Distribution

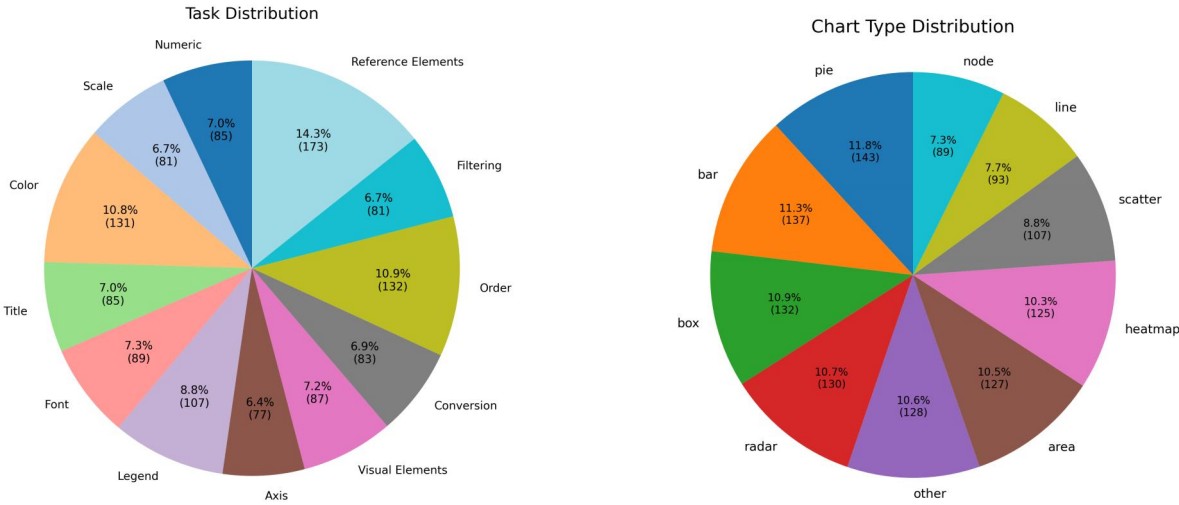

*Figure 2.* The distribution of chart types and editing tasks.

Figure 2 shows the distributions of chart types and editing tasks. The data are relatively well balanced, providing broad coverage of chart types and editing tasks.

## C. Stage-drop Statistics

*Table 10.* Step-drop Stastics.

| Stage | Images | Instructions | Reason | Filtering Rate |
|---|---|---|---|---|
| Data Collection | 10,000 | - | - | - |
| Diversity Filtering | 1,000 | - | CLIP+k-means | 90.0% |
| Chart-Code Pair Gen. | 997 | - | Code error | 0.3% |
| Inst. Generation | 997 | 1,457 | - | - |
| Manual Filtering | 808 | 1,211 | Overlap/Truncation/Invalid edit/Code or rendering failures | 18.2% (∼7% corrected) |

Table 10 shows the stage-drop statistics throughout the dataset construction process, highlighting the filtering rates and underlying reasons for each refinement step.

# D. Prompts Used in Experiments

## D.1. Prompts for Generating Edit Instruction

```
You are a professional Data Visualization Editing Instruction Generator and
Executor. Your task is to receive a Python chart code provided by the user.
You must operate under the assumption that you can only see the chart image
generated by the code and the list of editing categories. Your goal is to
select an editing task from the predefined categories, generate a clear
editing instruction, and then output the complete, runnable Python code that
executes that instruction.

Core Principle: Your selection of edits and instruction construction should
be based on how to enhance or alter the chart's visual presentation (as if
manipulating it through a graphical user interface), rather than strictly
relying on the underlying Python code structure.

### ASSIGNED TASK

Your task is EXCLUSIVELY to perform an edit within the following category:
- Main Category: {Main Category}
- Subcategory: {Subcategory}
- Task Description: {Description}

### REQUIREMENTS

1. Analyze the provided image and code.
2. Generate a specific editing instruction based on the "Task Description"
above (Example style: {example_inst}).
3. OUTPUT IMAGE: The generated Python code MUST save the resulting figure
using:
    plt.savefig('edit_image.png')
    Do NOT use plt.show(). Ensure all labels and titles are visible in the
    saved image.

### OUTPUT FORMAT

You must output:
1. The complete, updated Python code in a ```python ``` block.
2. A JSON object in a ```json ``` block with:
    - "file_id": "image_{main_cat}_{suffix}"
    - "selected_category": "{main_cat}"
    - "selected_subcategory": "{sub_name}"
    - "edit_instruction": "[Your generated specific instruction]"

### Model Response Example Structure

```python
import matplotlib.pyplot as plt\nimport numpy as np\n\n# ... (Original data
and plotting code)\n\n# EDIT OPERATION: Add Reference Line\nmean_value =
np.mean(data_values)\nax.axhline(mean_value, color='red', linestyle='--',
label=f'Mean Value: {{mean_value:.2f}}')\nax.legend()\n ... (Other plot
settings and plt.show())
```

```
```

```json
{{
  "file_id": "image_Global_Reference_Elements",
  "selected_category": "Global",
  "selected_subcategory": "Add/Modify Reference Elements",
  "edit_instruction": "Add a horizontal reference line representing the
  overall mean value of all data points. The line should be dashed and
  red.",
}}
```

## D.2. Prompts for Evaluation

```
You are a rigorous, impartial, and detail-oriented evaluator for chart image
editing tasks.
Your goal is to objectively assess whether a predicted chart image correctly
follows the given editing instruction while preserving irrelevant content.

You must strictly follow the scoring criteria and output format.
Do not provide explanations or reasoning.

## Evaluation Input

[Instruction]
<Instruction>

[Images]
<Image 1>: Original chart (before editing)
<Image 2>: Reference chart (after correct editing)
<Image 3>: Predicted chart (model output)

[Evaluation Task]
Evaluate the predicted chart image according to the instruction and the
reference images.

You must score the result on two independent dimensions:

### 1. Correctness (1 to 5)
Correctness measures whether the predicted chart correctly applies all edits
required by the instruction, including both visual-level modifications
(e.g., colors, styles, labels, legends, axes, titles, annotations) and
semantic or logical edits (e.g., data filtering, aggregation, sorting).

The evaluation should primarily compare the predicted chart with the
reference chart, while using the original chart to determine the scope of
instruction-required changes. Only chart elements that are explicitly
mentioned or logically implied by the instruction should be considered.

Scoring rules:
```

```
- 5: All required edits are fully correct and match the ground-truth image.
- 4: Most required edits are correct; only minor discrepancies.
- 3: Required edits are partially correct with noticeable errors.
- 2: Required edits are largely incorrect.
- 1: Required edits are missing or completely wrong.

### 2. Consistency (1 to 5)
Consistency measures whether the predicted chart preserves the intended
semantics and relationships of chart elements that are not affected by the
instruction, even if their visual appearance or layout changes due to
instruction-required logical edits.

Consistency should **not penalize global visual or structural changes** that
are a necessary consequence of correctly executing the instruction (e.g.,
filtering data points, re-sorting categories, or changing aggregation
levels).

Scoring rules:
- 5: All non-instructed elements remain unchanged.
- 4: Very minor unintended changes with negligible impact.
- 3: Some unintended but limited changes.
- 2: Many unintended changes that affect chart fidelity.
- 1: Extensive unintended changes; the chart is largely inconsistent with
the original.

## Output Format (Strict)

Return only the following JSON object:

```json
{
  "correctness": <integer from 1 to 5>,
  "consistency": <integer from 1 to 5>
}
```

Do not include explanations, reasoning steps, or any additional text.
```

## E. Detailed Definition and implementation of Metrics

We adopt a set of widely used image similarity and perceptual quality metrics to evaluate both low-level fidelity and high-level semantic consistency between a generated image $\hat{I}$ and its corresponding ground-truth image $I$, including Peak Signal-to-Noise Ratio (PSNR), Structural Similarity Index Measure (SSIM), Learned Perceptual Image Patch Similarity (LPIPS), CLIP similarity, and DINO similarity.

**PSNR** PSNR measures pixel-wise reconstruction quality based on the mean squared error (MSE), defined as

$$\text{MSE}(I, \hat{I}) = \frac{1}{N} \sum_{i=1}^{N} \left( I_i - \hat{I}_i \right)^2, \tag{14}$$

$$\text{PSNR}(I, \hat{I}) = 10 \log_{10}\left(\frac{L^2}{\text{MSE}(I, \hat{I})}\right), \qquad (15)$$

where $N$ denotes the total number of pixels and $L$ is the maximum possible pixel value. Higher PSNR indicates better pixel-level fidelity.

**SSIM**   SSIM evaluates perceptual similarity by jointly modeling luminance, contrast, and structural information:

$$\text{SSIM}(I, \hat{I}) = \frac{(2\mu_I \mu_{\hat{I}} + C_1)(2\sigma_{I\hat{I}} + C_2)}{(\mu_I^2 + \mu_{\hat{I}}^2 + C_1)(\sigma_I^2 + \sigma_{\hat{I}}^2 + C_2)}, \qquad (16)$$

where $\mu$, $\sigma^2$, and $\sigma_{I\hat{I}}$ denote the mean, variance, and covariance of the images, respectively, and $C_1, C_2$ are small constants for numerical stability. Higher SSIM values indicate stronger structural similarity.

**LPIPS**   LPIPS measures perceptual similarity using deep feature representations extracted from a pretrained network:

$$\text{LPIPS}(I, \hat{I}) = \sum_l \frac{1}{H_l W_l} \sum_{h,w} \left\| w_l \odot \left(\phi_l(I)_{hw} - \phi_l(\hat{I})_{hw}\right) \right\|_2^2, \qquad (17)$$

where $\phi_l(\cdot)$ denotes the feature activations from layer $l$, $H_l$ and $W_l$ are the spatial dimensions of the feature map, and $w_l$ is a learned channel-wise weighting vector. Lower LPIPS values indicate higher perceptual similarity.

**CLIP Similarity**   CLIP similarity evaluates semantic alignment in a joint vision-language embedding space. Given image embeddings $f_{\text{clip}}(\cdot)$ extracted by a pretrained CLIP image encoder, the similarity is computed as the cosine similarity:

$$\text{CLIP}(I, \hat{I}) = \frac{f_{\text{clip}}(I)^\top f_{\text{clip}}(\hat{I})}{\|f_{\text{clip}}(I)\|_2 \|f_{\text{clip}}(\hat{I})\|_2}. \qquad (18)$$

Higher CLIP similarity indicates better semantic consistency between images.

**DINO Similarity**   DINO similarity measures semantic similarity based on self-supervised visual representations. Using a pretrained DINO encoder $f_{\text{dino}}(\cdot)$, we compute

$$\text{DINO}(I, \hat{I}) = \frac{f_{\text{dino}}(I)^\top f_{\text{dino}}(\hat{I})}{\|f_{\text{dino}}(I)\|_2 \|f_{\text{dino}}(\hat{I})\|_2}. \qquad (19)$$

This metric captures high-level semantic similarity independent of pixel-level alignment.

**Implementation Details.**   For PSNR and SSIM, we use the default implementation provided by the `scikit-image` library, which computes SSIM over local windows with standard parameter settings. LPIPS is computed using the AlexNet-based model (`alex`) as proposed in the original work. For CLIP similarity, we adopt the pretrained CLIP image encoder `CLIP-ViT-L/14-336px`, and compute cosine similarity between the corresponding image embeddings. For DINO similarity, we use the pretrained `DINOv3-ViT-L/16-300M` model and similarly evaluate cosine similarity between the corresponding image embeddings.

## F. Comparison with Code-Mediated Editing

We compare end-to-end image-based chart editing models with a strong code-mediated baseline, GPT-5.1, which performs editing by first generating chart code and then rendering the result. This setting reflects a common alternative paradigm for chart editing and serves as a strong upper-bound baseline in terms of symbolic reasoning and program execution.

As shown in Table 11, GPT-5.1 achieves a high execution success rate (93.14%), indicating that code-mediated pipelines are reliable when the generated code can be correctly executed. However, despite this advantage, GPT-5.1 underperforms the best end-to-end image editing model (Nano Banana) across most objective metrics, including SSIM (0.77 vs. 0.83), PSNR (13.59 vs. 17.53), and LPIPS (0.42 vs. 0.23), as well as subjective Correctness and Consistency scores. This suggests that

*Table 11.* Comparison with GPT-5.1, a strong code-mediated image editing model, and the best end-to-end image editing model on Chart E$^3$.

| Model | Exec. (%) | Objective | | | | | Subjective | |
|---|---|---|---|---|---|---|---|---|
| | | SSIM | PSNR | CLIP | DINO | LPIPS↓ | Cor. | Con. |
| GPT-5.1 (w/ code) | 93.14 | 0.77 | 13.59 | 0.95 | 0.92 | 0.42 | 3.97 | 4.31 |
| GPT-Image-1.5 | - | 0.79 | 15.05 | **0.96** | 0.95 | 0.36 | 3.79 | 4.28 |
| Nano Banana | - | **0.83** | **17.53** | 0.96 | **0.96** | **0.23** | **4.18** | **4.65** |

translating visual edits into executable code does not necessarily guarantee higher perceptual fidelity or better preservation of non-edited chart elements.

We note that the two paradigms are not strictly equivalent: code-mediated editing benefits from explicit program structures but relies on intermediate representations that may diverge from the original chart appearance, while end-to-end models operate directly in the image domain and are optimized for perceptual alignment. The results indicate that, even when compared against a powerful code-based model, end-to-end image editing can achieve superior visual quality and more faithful execution of editing intents, particularly for complex or visually sensitive chart edits.

## G. Error Analysis

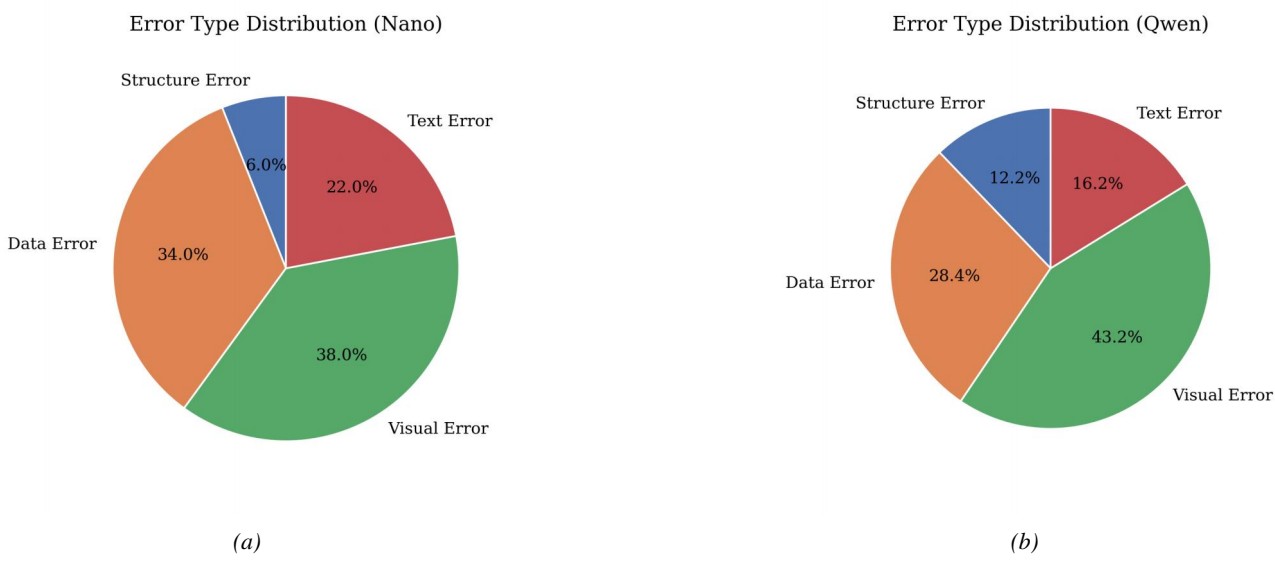

*(a)*        *(b)*

*Figure 3.* The error analysis of Nano Banana and Qwen-Image-Edit.

We manually conduct an error analysis to better understand the failure modes of current end-to-end chart editing models. Errors are categorized into four types: (1) Structure Errors, where severe rendering deviations or chart layout corruption result in invalid or broken charts; (2) Visual Errors, including incorrect colors, misplaced visual elements, or inconsistent styles; (3)Data Errors, where edited charts contain incorrect values, ordering, or filtering results; and (4) Text Errors, such as incorrect titles, labels, or legends. We analyze Nano Banana and Qwen-Image-Edit as representative closed- and open-source models, respectively. As shown in Figure 3, visual errors dominate for both models, accounting for 38.0% of failures in Nano Banana and 43.2% in Qwen-Image-Edit, indicating persistent challenges in precise control of fine-grained visual attributes. Data errors are the second most frequent (34.0% and 28.4%), reflecting difficulties in executing data-dependent edits while preserving chart correctness. Text errors are less common, particularly for Qwen-Image-Edit (16.2%) compared to Nano Banana (22.0%), suggesting relatively more stable text generation but limited overall reliability. Structure errors occur least frequently (6.0% and 12.2%) but are the most severe, often leading to invalid or broken charts.

Overall, Nano Banana shows stronger structural robustness, while both models exhibit similar dominant failure patterns, with visual and data errors accounting for the majority of failures. These results highlight that chart editing remains bottlenecked

by precise execution and structural consistency beyond semantic understanding alone.

## H. Case Study

Figure 4, Figure 5, Figure 6 and Figure 7 present representative examples for qualitative case studies.

## I. Limitation

Despite its strengths, ChartE$^3$ has several limitations. The current dataset scale, while sufficient for systematic evaluation, is still modest compared to large-scale multimodal corpora, which may limit coverage of rare chart styles and long-tail editing behaviors. In addition, parts of the data construction rely on proprietary APIs, potentially affecting reproducibility and accessibility. ChartE$^3$ may also exhibit language and regional biases, as the data is primarily collected from dominant languages and regions. Finally, the benchmark focuses exclusively on static two-dimensional charts and does not cover interactive visualizations such as dashboards, animated charts, or geographic maps. Future work will expand ChartE$^3$ in scale and diversity, reduce reliance on proprietary APIs, and extend evaluation beyond static 2D charts to interactive and dynamic visualizations such as dashboards and maps.

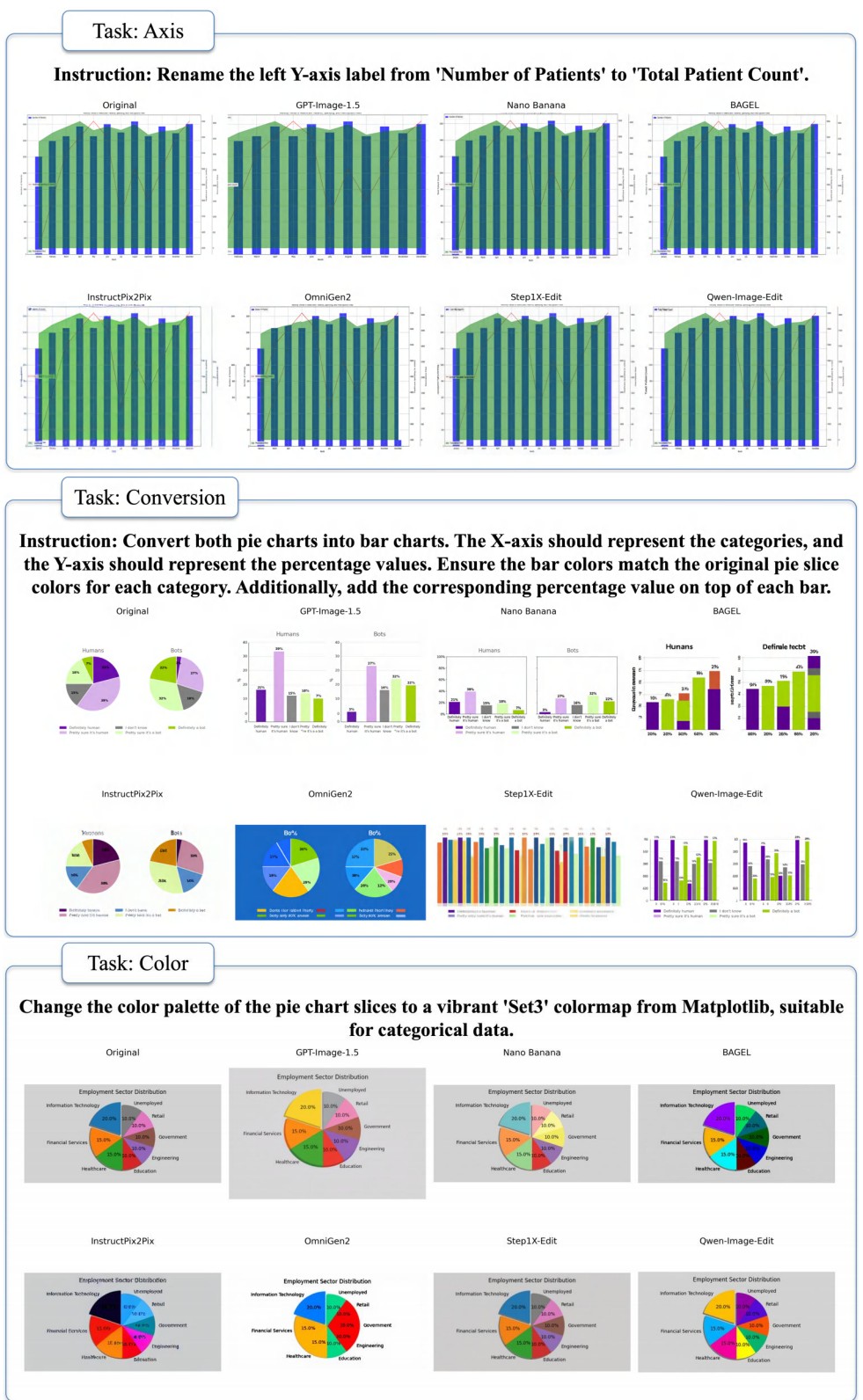

*Figure 4.* Editing categories: Axis, Conversion, and Color.

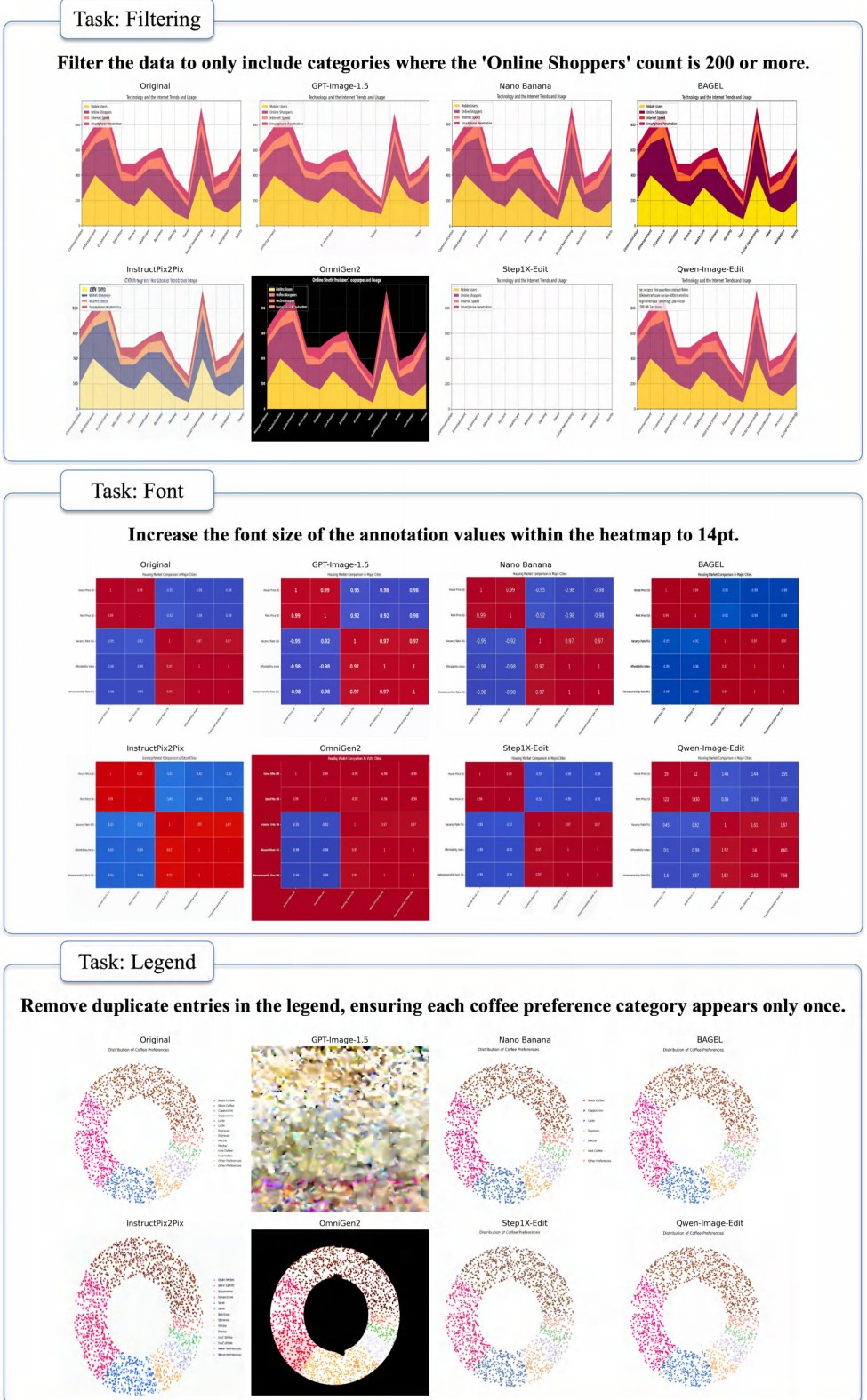

*Figure 5.* Editing categories: Filtering, Font, and Legend.

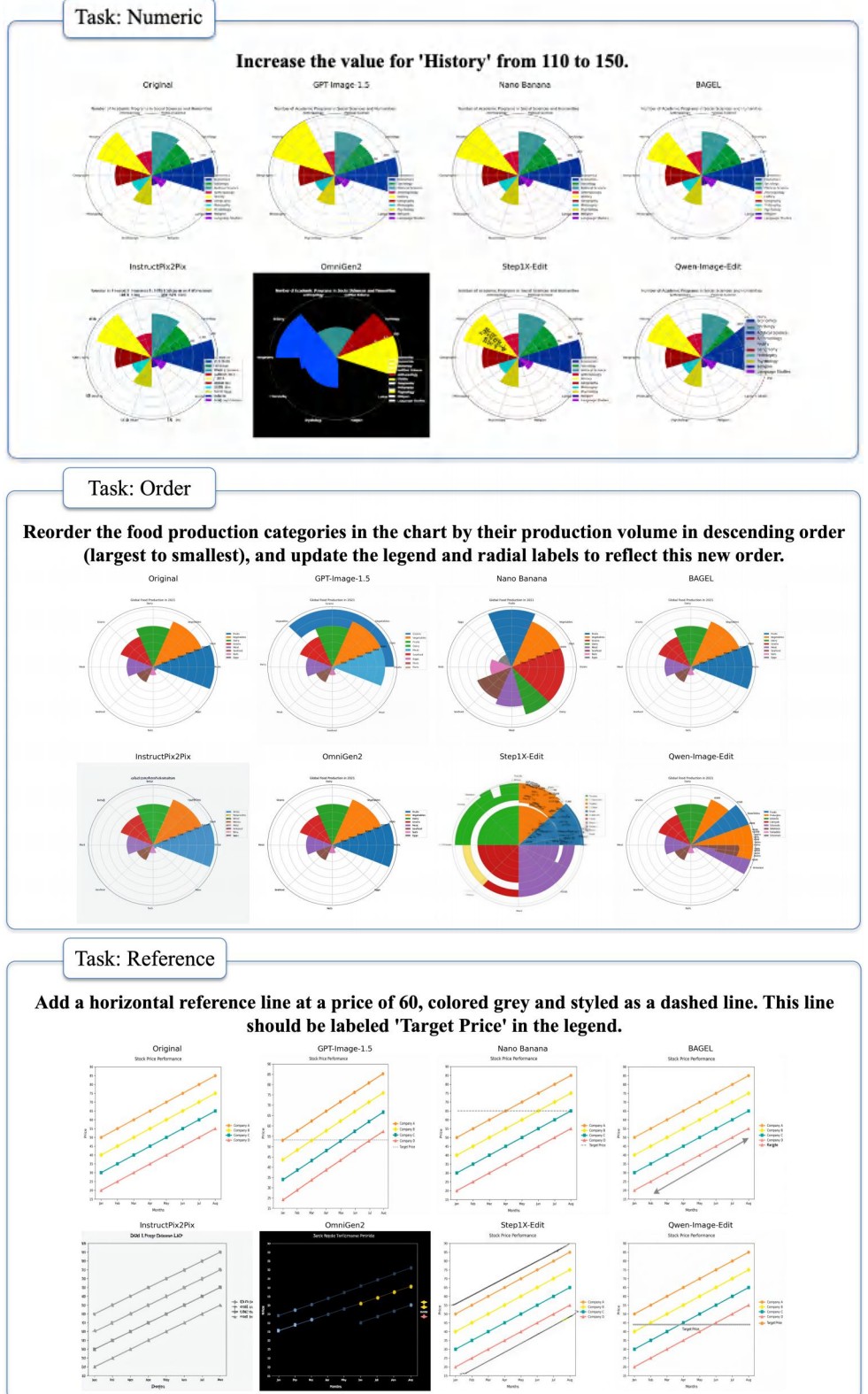

*Figure 6.* Editing categories: Numeric, Order, and Reference.

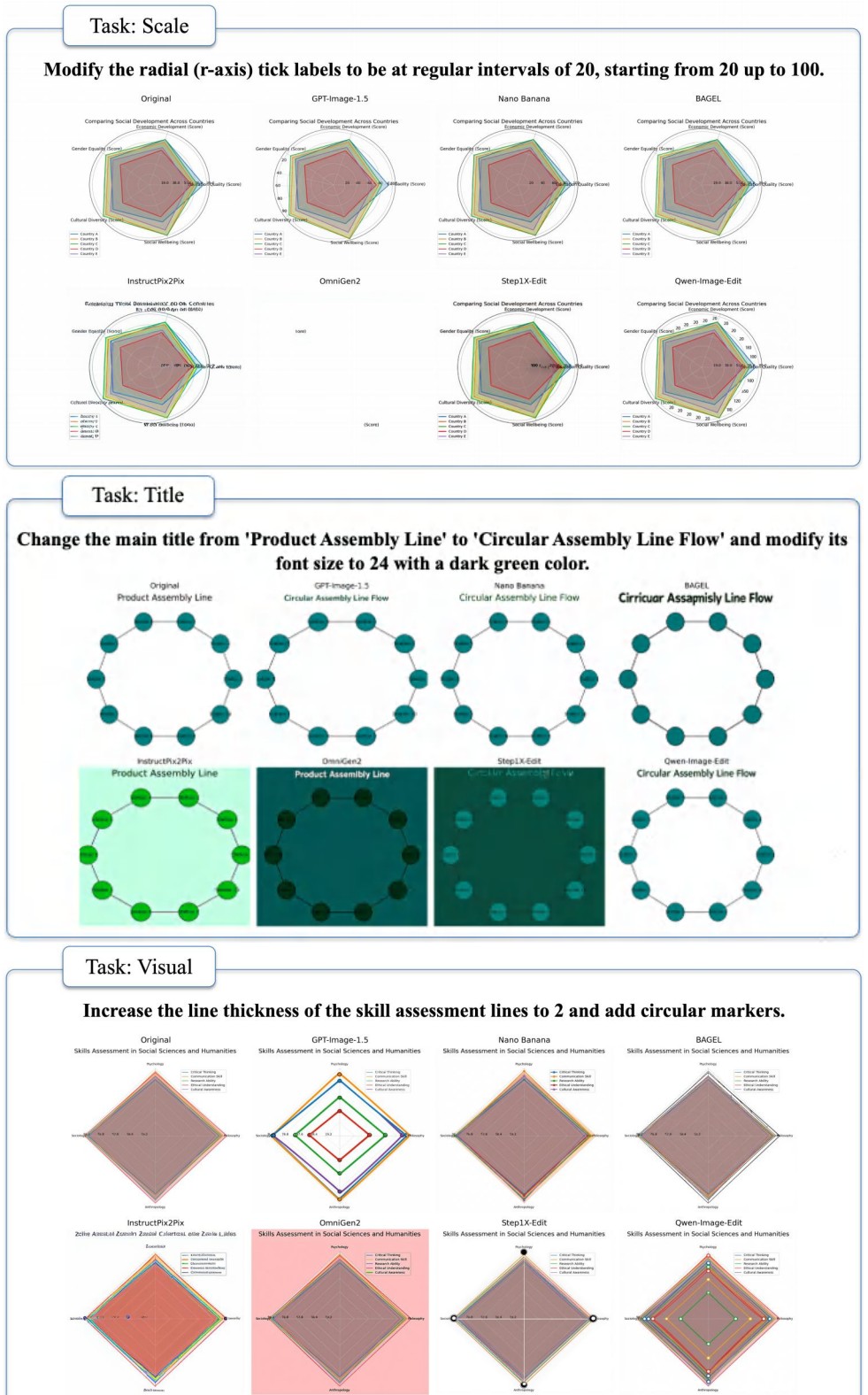

*Figure 7.* Editing categories: Scale, Title, and Visual.

