# OpenReview forum: "ChartE$^{3}$: A Comprehensive Benchmark for End-to-End Chart Editing"
_ICML.cc/2026/Conference — ICML 2026 regular_

### Official Review · Reviewer_C3S7 · 2026-03-02

**Soundness:** 2
**Presentation:** 3
**Significance:** 2
**Originality:** 3
**Overall Recommendation:** 4
**Confidence:** 5

**Summary:**

Current chart-to-code or chart editing approaches rely on converting images into code as an intermediate step before rendering edits, which primarily tests coding ability rather than true end-to-end visual editing.

This paper introduces ChartE3, an end-to-end chart editing benchmark that evaluates models directly on image-based editing without intermediate code or natural language supervision. It includes 1,200 human-curated samples across local and global editing tasks. Benchmark results show significant performance gaps in state-of-the-art multimodal models, especially for global edits, exposing clear limitations in current end-to-end chart editing capabilities.

**Compliance With Llm Reviewing Policy:**

Affirmed.

**Final Justification:**

The authors’ rebuttal has addressed my main concerns. Their clarifications and additional explanations alleviate my earlier doubts to a satisfactory extent. I therefore lean towards positive assessment of the paper.

**Key Questions For Authors:**

See weaknesses above.

Other question are:

1: It is also unclear what the practical advantage of an image-based chart editing approach is compared to a code-based one. Code-based chart editing is typically more interactive, reusable, and easier to refine according to user needs. In contrast, image-based editing may offer less flexibility, making it harder for users to further modify or adjust the generated result if they are not satisfied.

2: Sample counts per task/type and stage-drop statistics: Could you provide exact sample counts per task (for Table 6) and chart type (for Table 7), as well as statistics on how many candidate charts are discarded at each step of the pipeline (CLIP+k-means filtering, code reconstruction failures, manual filtering)? This would help assess representativeness and potential selection bias.

**Limitations:**

see above weakness and questions

**Strengths And Weaknesses:**

Strength:

1: Clear problem focus and benchmark definition. The paper carefully distinguishes code-mediated chart editing from the proposed end-to-end image editing formulation, and positions ChartE³ as targeting the latter, which is currently under-explored relative to chartEdit benchmarks.

2: Hybrid evaluation design with validation of GPT judge. The combination of SSIM, PSNR, CLIP, DINO, and LPIPS with GPT-based correctness and consistency scores is appropriate. It also provide evidence that GPT scores correlate well with human rankings, strengthening the credibility of the subjective evaluation.

3: Insightful breakdown analyses. Beyond the global results in Table 5, the per-task (Table 6) and per-chart-type (Table 7) analyses provide useful insights.

Weaknesses:

1: Missing chart-specific baselines: There is no direct comparison with a chart-specialized, code-mediated pipeline adapted to image generation

2: Modest dataset scale for a benchmark: 808 images and 1,211 instructions are decent but not large by modern benchmark standards, especially when spread over 10 chart types and 12 tasks.

3: Insufficient reporting of construction-stage filtering and failures: While Section 3.2 describes the CLIP + k-means filtering and iterative code reconstruction, the paper does not quantify how many samples are discarded at each step or how many require manual correction.

4: Evaluation choices and metrics are not fully interrogated: The paper states there is a “strong consistency between objective and subjective evaluations” (Section 4.2), but provides no quantitative correlation analysis (e.g., Pearson/Spearman correlation between CLIP/SSIM and GPT scores across images). Also, objective metrics are computed after resizing images, while GPT judges use original resolution. This may cause some misalignment, which is not discussed.

---

> ### Author Rebuttal · Authors · 2026-03-30
>
> We thank the reviewers for their comments and reply as follows:
>
> > Response to W1: Missing chart-specific baselines
>
> We compare our pipeline with recent chart-specialized, code-mediated Benchmark, ChartM$^{3}$ and ChartEdit, as shown in follow:
>
> | Benchmark | Data collection | Diversity Filtering | Chart-Code Pair Gen. | Inst. Gen. | Manual Filtering |
> | :--- | :--- | :---: | :---: | :---: | :---: |
> | ChartM$^{3}$ | Syn. | N | Y | Y | N |
> | ChartEdit | Real | N | Y | Y | Part |
> | Ours. | Syn. & Real | Y | Y | Y | Y |
>
> Our pipeline subsumes prior works by **incorporating all existing components while additionally introducing diversity filtering and thorough manual filtering**. Unlike prior datasets that **rely on a single source** (e.g., arXiv charts), we combine both synthetic and real data from multiple sources to enhance diversity. We also observe that existing datasets often suffer from **textual or visual occlusions that degrade data quality**; in contrast, our manual verification ensures more reliable samples. Further differences in chart and task types are discussed in Table 1 in the paper.
>
> > Response to W2: Modest dataset scale for a benchmark
>
> - Firstly, our dataset size is carefully designed to be **consistent with recent chart editing and general image generation/editing benchmarks** as follow table:
>
> | Benchmark       | Type              | Image | Inst. | Avg. Inst. per task |
> |:---|:---:|:---:|:---:|:---:|
> | ChartM$^{3}$    | Chart Editing     | 1000  | 1000        | ~100                |
> | ChartEdit       | Chart Editing     | 233   | 1405        | ~200                |
> | GenEval         | Image Gen.  | /     | 553         | ~100                |
> | DPG             | Image Gen.  | /     | 1000        | ~200                |
> | GEdit-Bench     | Image Editing     | 606   | 606         | ~50                 |
> | ImageEdit-Bench | Image Editing     | 808   | 808         | ~60                 |
> | Ours.           | E2E Chart Editing | 808   | 1211        | ~100                |
>
> Benchmarks such as **GenEval, DPG and ImageEdit-Bench are widely used standards**, and our scale is well aligned with these representative works.
>
> - Secondly, the scale is optimized for practical reproducibility. Image Gen./Editing involves relies on **costly APIs (e.g., Nano Banana), typically ranging from \$0.1-0.8 per image**. Overly large benchmarks would hinder accessibility and iterative evaluation.
>
> - Finally, unlike large-scale automated benchmarks, all instructions are curated with diversity filtering and expert verification, reducing bias and annotation errors. This ensures more reliable evaluation.
>
> > Response to W3 & Q2: Sample counts per task/type and stage-drop statistics.
>
> - Firstly, we give the task/type distribution and will provide sample counts in Appendix B, due to the rebuttal character limit.
>
> - Secondly, we provide the requested quantification of our data construction pipeline:
>
> | Stage | Images | Inst. | Reason | Filtering Rate |
> | :--- | :---: | :---: | :--- | :---: |
> | Data Collection | 10,000 | - | - | - |
> | Diversity Filtering | 1,000 | - | CLIP+k-means | 90.0% |
> | Chart-Code Pair Gen. | 997 | - | Code error | 0.3% |
> | Inst. Generation | 997 | 1,457 | - | - |
> | Manual Filtering | 808 | 1,211 | Overlap/truncation/Invalid edit/Code or rendering failures | 18.2% (~7% corrected) |
>
> We will include this result in the next version to provide more transparency.
>
> > Response to W4: Evaluation choices and metrics are not fully interrogated.
>
> We provide the Spearman/Pearson correlation between objective metrics and GPT scores. The **high correlation** empirically supports our evaluation framework.
>
> | | SSIM | PSNR | CLIP | DINO | LPIPS$\downarrow$ |
> | :--- | :---: | :---: | :---: | :---: | :---: |
> | **Cor.** | 0.83/0.46 | 0.88/0.56 | 0.98/0.77 | 0.99/0.79 | -0.74/-0.58 |
> | **Con.** | 0.69/0.53 | 0.77/0.62 | 0.87/0.84 | 0.93/0.86 | -0.64/-0.68 |
>
> While objective metrics require resizing, the GPT Judge serves as a vital semantic complement, especially for text recognition. Our tests show that GPT scores are highly robust to resolution changes (1.02$\times$ and 0.98$\times$), **confirming that resizing misalignment does not compromise reliability**.
>
> | Condition | Cor. | Con. |
> | :--- | :---: | :---: |
> | Origin | 4.18 | 4.65 |
> | Enlarge | 4.20 | 4.65 |
> | Shrink | 4.21 | 4.65 |
>
> > Response to Q1: unclear the practical advantage.
>
> We understand the concern and clarify it from two aspects:
>
> - **Practical value.** End-to-End Image editing lowers the barrier for **non-expert users who are unfamiliar with chart specification languages**, and is particularly useful when the original chart code is unavailable.
>
> - **Academic value.** This setting directly evaluates a model’s visual understanding, reasoning, and consistency, without relying on structured code as an intermediate. Moreover, many image generation models do not produce executable chart code, making image-based evaluation more general and model-agnostic.

---

> > ### Author Rebuttal · Reviewer_C3S7 · 2026-04-01
> >
> > The authors have addressed my major concerns, and I would like to raise my score accordingly.

---

> > > ### Author Response · Authors · 2026-04-03
> > >
> > > We sincerely thank the reviewer for the valuable comments and efforts. We are glad that the concerns have been fully resolved.

---

### Official Review · Reviewer_sUTh · 2026-03-09

**Soundness:** 3
**Presentation:** 3
**Significance:** 2
**Originality:** 2
**Overall Recommendation:** 4
**Confidence:** 3

**Summary:**

The paper introduces ChartE3, a new benchmark designed to evaluate the end-to-end chart editing capabilities of multimodal large language models (MLLMs). Unlike existing approaches that rely on intermediate code generation or program representations, this benchmark frames chart editing as a direct image-to-image visual transformation task. ChartE3 contains over 1,200 high-quality, human-curated samples spanning 10 chart types and 12 distinct editing tasks, partitioned into local (appearance-level) and global (data-centric) edits. The authors evaluate several state-of-the-art open-source and closed-source models using a hybrid protocol of objective similarity metrics (e.g., SSIM, LPIPS) and subjective GPT-based evaluation. The benchmarking reveals that while closed-source models lead in performance, all models struggle significantly with global, data-centric edits compared to localized appearance modifications.

**Compliance With Llm Reviewing Policy:**

Affirmed.

**Final Justification:**

My concerns are addressed by the rebuttal; I will keep my positive score.

**Key Questions For Authors:**

- Given that end-to-end image generation can sometimes introduce minor, semantically irrelevant layout shifts (e.g., shifting a legend a few pixels), how sensitive are objective metrics like SSIM and PSNR to these acceptable variations?

- Could you provide more specific examples of the criteria human experts used during the manual filtering stage to discard samples with ambiguous editing outcomes?

- In Appendix E, you compare the end-to-end models against a code-mediated baseline (GPT-5.1) that has access to the ground-truth code. To establish a stricter "vision-only" baseline, could you consider a full "image-to-code" followed by "code-to-image" (rendering) pipeline where the model can first extract the code purely from the input image before editing it?

**Limitations:**

yes

**Strengths And Weaknesses:**

### Strengths

- Delicated Data Pipeline: The dataset is constructed via a well-thought-out, five-stage pipeline that includes diversity filtering, reflection-style code generation for annotation, and manual verification by human experts. Covering both fine-grained local edits (e.g., fonts, colors) and holistic global edits (e.g., data filtering, type conversion) ensures a thorough evaluation landscape.

- Comprehensive Evaluation: The authors evaluate 7 models on 12 distinct editing tasks by combining standard pixel-level and perceptual metrics with subjective GPT-based scoring for correctness and consistency. Also, the paper empirically validates that the GPT-based subjective scores strongly align with human judgments.

### Weaknesses

- Limitations of the Evaluation Framework: General image metrics like PSNR and SSIM can be misleading for charts. Minor pixel shifts that are completely imperceptible to humans can drastically alter the metric scores. Because the benchmark lacks deterministic OCR data extraction metrics to verify text accuracy, it relies entirely on GPT-5.1 to evaluate semantic Correctness and Consistency. Anchoring the primary evaluation to a closed-source, continuously updating API introduces significant reproducibility and drift concerns for a long-term standard.

- Depth of Error Analysis: The paper provides a high-level error analysis categorized into Structure, Visual, Data, and Text errors. However, it lacks a deeper investigation into why models fail so drastically on global data edits while succeeding at additive global edits (like reference lines). Further mechanistic insights would elevate the analysis.

---

> ### Author Rebuttal · Authors · 2026-03-30
>
> We are glad the reviewer finds our proposed benchmark dedicated and comprehensive. We respond to the reviewer's questions below.
>
> > Response to W1 \& Q1: Limitations of the Evaluation Framework.
>
> - The impact of Minor Pixel Shifts on objective metrics.
>
> We acknowledge that pixel-level metrics like PSNR and SSIM can be sensitive to minor shifts. However, our empirical sensitivity analysis on the ChartE$^{3}$ dataset demonstrates that these metrics remain reasonably stable under slight perturbations. Even with Gaussian noise (std=0.3) or pixel shifts (within 10px), the fluctuations in scores are acceptable:
>
> | Condition | SSIM | PSNR | CLIP | DINO | LPIPS |
> | :--- | :---: | :---: | :---: | :---: | :---: |
> | Origin | 0.83 | 17.53 | 0.96 | 0.96 | 0.23 |
> | w/ Gaussian Noise | 0.83 | 17.53 | 0.95 | 0.96 | 0.24 |
> | w/ 10px Shift | 0.81 | 16.22 | 0.95 | 0.96 | 0.25 |
>
> The data shows that **PSNR only dropped by ~7% and SSIM by ~2% under a 10px shift**, which does not "drastically alter" the evaluation outcome.
>
> To mitigate the limitations of pixel-level metrics, we incorporated CLIP and DINO, which focus on high-level semantic features and showed almost zero fluctuation during our tests. Our framework **does not rely on a single metric**; instead, **it combines perceptual, semantic and LLM-based subjective evaluations** to ensure a comprehensive and reliable assessment of chart editing quality.
>
> - Analysis of OCR metric.
>
> We thank the reviewers for their and further introduce a deterministic OCR-based metric to assess text accuracy. Specifically, we use EasyOCR to extract text and compute **Normalized Edit Distance (NED)** between predictions and references.
>
>
> | Model          | NED ↑ |
> |----------------|--------|
> | GPT-Image-1.5  | 0.7379 |
> | Nano Banana    | **0.8038** |
> | BAGEL          | 0.7366 |
> | InstructPix2Pix| 0.2294 |
> | Step1X-Edit    | 0.6491 |
> | OmniGen2       | 0.5172 |
> | Qwen-Image     | 0.7262 |
>
> The results show consistent trends with our original metrics, where Nano Banana performs best overall, and BAGEL and Qwen-Image are the strongest among open-source models. This consistency suggests that GPT-based evaluation is reliable, as it aligns well with independent, text-level verification.
>
> - Analysis of API-based evaluation
>
> We understand the reviewer's concerns. Using API-based evaluators is a common practice in recent benchmarks, as they provide strong and scalable semantic assessment. To mitigate reproducibility and drift issues, we additionally introduce these objective metrics as a complementary evaluation.
>
> Importantly, we observe consistent trends between GPT-based and objective metrics, supporting the reliability of our results. For long-term usage, we plan to update results as the API evolves, ensuring the benchmark remains current.
>
> > Response to W2: Depth of Error Analysis.
>
> To provide mechanistic insights, we quantified the structural change across four global tasks using SSIM between original and edited images:
>
> | Task Category | SSIM |
> | :--- | :---: |
> | Conversion | 0.76 |
> | Order | 0.84 |
> | Filtering | 0.84 |
> | **Reference Elements** | **0.91** |
>
> We suppose that the performance disparity stems from the degree of structural preservation. Reference Elements (e.g., adding lines) requires complex reasoning but involves minimal pixel-level modification (SSIM: 0.91), allowing models to maintain global consistency. In contrast, Conversion or Order tasks require drastic global layout restructuring (lower SSIM), which likely exceeds the current generative stability of MLLMs, leading to more frequent failures.
>
>
> > Response to Q2: Provide more specific examples of the criteria human experts used.
>
> We thank the reviewer for the question. During the manual filtering stage, human experts removed samples based on the following criteria:
>
> - Severe overlap/truncation between text and visual elements leading to illegible or ambiguous outputs (the most common case), e.g., labels overlapping with bars/lines, legends covering key regions, or text rendered on top of data points.
>
> - Invalid or meaningless edits, e.g., adding axes to chart types where they are not applicable (such as node-based diagrams), inserting irrelevant annotations, or modifying elements that do not exist in the original chart.
>
> - Code compilation or rendering failures, including issues caused by deprecated libraries or incorrect plotting code, e.g., syntax errors, unsupported parameters, or incomplete rendering of chart components.
>
> > Response to Q3: could you consider a full "image-to-code" followed by "code-to-image" (rendering) pipeline?
>
> We apologize for the confusion. The code-mediated baseline in Appendix E does not have access to the ground-truth code. Instead, it follows the pipeline as you say: image → code → edited code → rendered image. Specifically, the model first infers the chart code from the input image, and then performs editing based on the generated code. We will provide a clearer narrative to prevent confusion.

---

> > ### Author Rebuttal · Reviewer_sUTh · 2026-04-01
> >
> > Thanks to the authors for the rebuttal. My concerns are addressed; I will keep my positive score.

---

> > > ### Author Response · Authors · 2026-04-03
> > >
> > > We sincerely thank the reviewer for the valuable comments and efforts. We are glad that the concerns have been fully resolved.

---

### Official Review · Reviewer_2sh9 · 2026-03-09

**Soundness:** 3
**Presentation:** 4
**Significance:** 4
**Originality:** 3
**Overall Recommendation:** 4
**Confidence:** 3

**Summary:**

This paper presents ChartE³, a new benchmark designed to test how well multimodal large language models can edit charts end-to-end. Instead of relying on intermediate code generation—which is the standard approach in recent literature—this work treats chart editing as a direct image-to-image task guided by natural language instructions. The authors curated a dataset of over 1,200 triplets consisting of an image, its code, and an editing instruction. The benchmark covers 10 chart types and 12 specific editing tasks. These tasks range from local cosmetic changes, like colors and fonts, to global structural edits, like data filtering or changing the chart type. Based on a mix of objective metrics and GPT-evaluated subjective scores, the experiments demonstrate that while top proprietary models handle local edits decently, all evaluated models still struggle significantly with global, data-centric modifications.

**Compliance With Llm Reviewing Policy:**

Affirmed.

**Final Justification:**

The rebuttal adequately addresses my concerns regarding dataset scale, evaluation reliability, and potential bias, leading me to maintain my positive recommendation.

**Key Questions For Authors:**

1. Given that your benchmark covers 10 chart types and 12 distinct editing tasks (totaling 120 unique combinations) using only 1,211 instructions, the average number of samples per condition is roughly 10. Does this small sample size provide enough statistical power to draw definitive conclusions about a model's performance on specific difficult tasks, such as data filtering or chart conversion?
2. Did you check the self-preference bias when using the GPT judge to evaluate outputs from GPT-Image-1.5?
3. Can you provide a granular breakdown demonstrating that the GPT judge maintains high human alignment specifically on the challenging "Global" editing tasks, or is the overall score artificially skewed by the easier "Local" edits?

**Limitations:**

yes

**Strengths And Weaknesses:**

Strengths:
- The paper is generally well written and clearly explains the limitations of existing code-to-chart approaches. The figures, such as the pipeline overview and error distribution plots, are helpful.
- The five-stage pipeline used to build the dataset is well designed. Feature-based sampling helps ensure visual diversity, and the final filtering by human experts improves the overall quality of the benchmark.
- The evaluation combines objective perceptual metrics such as SSIM and LPIPS with GPT-based scoring for correctness and consistency, which provides a more complete assessment.

Weaknesses:
- The dataset contains only 808 images, which is much smaller than existing chart benchmarks such as ChartQA or ChartX. This makes it harder to evaluate model generalization or performance on complex cases.
- The evaluation relies on a GPT model for subjective scoring. Since GPT-Image-1.5 is also one of the evaluated models, this may introduce self-preference bias.
- The GPT-based judge is validated on only 140 samples across many task types. This scale is relatively small and may not be sufficient to demonstrate reliability, especially for complex editing tasks.

---

> ### Author Rebuttal · Authors · 2026-03-30
>
> We are pleased and grateful to the reviewers for their recognition of our work. We reply to the concerns of the authors as follows.
>
> > Response to W1: The dataset contains only 808 images.
>
> We appreciate the reviewer’s concern regarding the dataset scale.
>
> - Firstly, we would like to clarify that ChartE$^{3}$ is a benchmark for End-to-End Chart Editing, which follows a different data paradigm from VQA tasks like ChartQA. Our dataset size is carefully designed to be **consistent with recent chart editing and general image generation/editing benchmarks** as follow table:
>
> | Benchmark       | Type              | Image | Instruction | Avg. Inst. per task |
> |-----------------|-------------------|-------|-------------|---------------------|
> | ChartM$^{3}$    | Chart Editing     | 1000  | 1000        | ~100                |
> | ChartEdit       | Chart Editing     | 233   | 1405        | ~200                |
> | GenEval         | Image Generation  | /     | 553         | ~100                |
> | DPG             | Image Generation  | /     | 1000        | ~200                |
> | GEdit-Bench     | Image Editing     | 606   | 606         | ~50                 |
> | ImageEdit-Bench | Image Editing     | 808   | 808         | ~60                 |
> | Ours.           | E2E Chart Editing | 808   | 1211        | ~100                |
>
> Benchmarks like GenEval, DPG, GEdit-Bench, and ImageEdit-Bench are among the **most frequently used standards** in the generative and image editing communities. As shown above, our scale (808 images, 1,211 instructions) is highly aligned with these representative works.
>
> - Secondly, the scale of ChartE$^{3}$ is optimized for practical reproducibility. Unlike text-based VQA, instruction-based chart editing involves proprietary image generation APIs (e.g., GPT-Image, NanoBanana) with **high unit costs**, typically ranging from \$0.1-0.8 per image. A prohibitively large dataset would make full-benchmark evaluation inaccessible to most academic labs, stifling secondary development.
>
> - Finally, **unlike large-scale automated benchmarks that often suffer from stylistic bias, and overlap elements, each of the 1,211 instructions in ChartE$^{3}$ has undergone strict diversity filtering and human verification**. This human-in-the-loop approach ensures each sample targets non-trivial challenges, leading to more reliable and definitive conclusions regarding MLLM capabilities.
>
> > Response to W2 \& Q2: Concerns on self-preference bias.
>
> To address potential self-preference bias, we conducted a human audit to verify the GPT Judge's alignment with expert human judgment across outputs from GPT-Image-1.5 and Nano Banana. We measured the Human Approval Rating (percentage of LLM-assigned scores corroborated by experts):
>
> | Metric | Human Approval Rating |
> | :--- | :---: |
> | Correctness (Cor.) | 81% |
> | Consistency (Con.) | 89% |
>
> The high agreement (**81% for Cor. and 89% for Con.**) demonstrates that the GPT Judge provides objective, criteria-based evaluations rather than showing a systematic bias toward models of the same family.
>
> In addition, we give stronger human validation in our **reply to w3**, and these further justify our conclusions.
>
> > Response to W3 \& Q3: Provide a granular breakdown demonstrating that the GPT judge.
>
> Following your suggestion, we conduct **more human-aligned validation** on 280 samples, consisting of 140 global and 140 local tasks. As shown below, LLM-based judgments achieve strong agreement with human preferences across both correctness and consistency metrics. These results support the **reliability of our evaluation framework**.
>
> | Task Type  | Metric        | Correctness | Consistency |
> |------------|---------------|-------------|-------------|
> | **Overall**| Pairwise Acc  |   0.785     |   0.867     |
> |            | NDCG@7        |   0.924     |   0.962     |
> | **Global** | Pairwise Acc  |   0.826     |   0.887     |
> |            | NDCG@7        |   0.941     |   0.957     |
> | **Local**  | Pairwise Acc  |   0.766     |   0.874     |
> |            | NDCG@7        |   0.919     |   0.971     |
>
> > Response to Q1: Concerns on the average number of samples per condition.
>
> We thank the reviewer for the insightful comment. The ~10 samples correspond to the **intersection of chart types and editing tasks (10 × 12)**, which is not the level at which we draw conclusions. Instead, our analysis is only conducted along **single dimensions**, with ~100 samples per task and ~120 per chart type, providing more reliable estimates.
>
> As shown in **Response to W1**, this setup aligns with common practices in image generation \& editing benchmarks, where evaluations are typically performed at the task level.
>
> In future work, we plan to expand the dataset and perform more fine-grained analysis over cross-combinations.

---

> > ### Author Rebuttal · Reviewer_2sh9 · 2026-04-01
> >
> > Thank you for the explanation provided by the authors. My concerns have been mainly addressed, and thus I will keep my positive score.

---

> > > ### Author Response · Authors · 2026-04-03
> > >
> > > We sincerely thank the reviewer for the valuable comments and efforts. We are glad that the concerns have been fully addressed.

---

### Official Review · Reviewer_GBBG · 2026-03-13

**Soundness:** 3
**Presentation:** 3
**Significance:** 3
**Originality:** 4
**Overall Recommendation:** 4
**Confidence:** 3

**Summary:**

This paper introduces ChartE3, a benchmark designed to evaluate end-to-end chart editing capabilities of multimodal models without relying on intermediate natural language programs or code generation. The benchmark focuses on two editing dimensions: local editing, involving appearance changes such as color or font modifications, and global editing, involving structural transformations such as data filtering or trend line addition. The dataset contains over 1,200 curated samples, each consisting of a chart image, the underlying chart code, and a multimodal editing instruction. The authors evaluate several state-of-the-art multimodal models and show that current systems perform reasonably well on local appearance edits but struggle substantially with global data-level edits, highlighting limitations in current chart editing capabilities.

**Compliance With Llm Reviewing Policy:**

Affirmed.

**Key Questions For Authors:**

To what extent does the benchmark rely on intermediate chart code when constructing ground-truth edits, and could this introduce bias into evaluation?
How robust are the subjective evaluations performed by LLM judges compared to human annotators?
Do failures on global editing primarily arise from visual reasoning limitations or from text rendering limitations in current models?

**Limitations:**

yes

**Strengths And Weaknesses:**

[Strengths]
Soundness
The dataset construction pipeline is clearly described and appears systematic, combining automatic generation with human curation.
The distinction between local and global editing tasks is well motivated and provides a useful diagnostic lens.
The benchmark includes both objective metrics and subjective evaluation, providing complementary perspectives.

Presentation
The paper is clearly structured and easy to follow.
Task taxonomy and evaluation setup are well explained, making it easy to understand how the benchmark differs from prior work.

Significance
Chart editing is an increasingly relevant task for multimodal systems, especially in data analysis and visualization workflows.
The benchmark may facilitate future research on structure-aware visual editing and multimodal reasoning.

Originality
The benchmark introduces a useful problem framing: evaluating chart editing directly from images rather than through intermediate code generation.
The explicit separation of local vs. global edits offers a useful perspective for diagnosing model weaknesses.


[Weaknesses]
Soundness
The evaluation partially depends on generated chart code to produce ground-truth edits, which somewhat weakens the claim of being fully “code-independent.”
The subjective evaluation relies heavily on automated LLM judging; stronger human validation would improve reliability.

Presentation
While the dataset construction is described clearly, reproducibility could be improved with more explicit details about prompt generation and filtering heuristics.

Significance
Although chart editing is an important niche problem, the broader machine learning impact may be somewhat limited compared to method-focused contributions.

Originality
The main contribution lies in benchmark design rather than algorithmic innovation.

---

> ### Author Rebuttal · Authors · 2026-03-30
>
> We thank the reviewers for their endorsement of our proposed benchmark and we reply to the reviewer's questions as follows.
>
> > Response to W1: Soundness.
>
> We thank the reviewer for the insightful comments.
>
> - On code-generated reference charts.
>
> We use programmatic generation to construct reference charts for **precise and controllable** annotation of edits (e.g., data, layout, style), which is critical for avoiding ambiguity in chart editing evaluation. Importantly, the evaluation is conducted purely on rendered images and extracted semantics, without requiring access to the underlying code. Thus, our setting remains code-independent at evaluation time.
>
> - Stronger human validation.
>
> Following your suggestion, we conduct **more human-aligned validation** on 280 samples, consisting of 140 global and 140 local tasks. As shown below, LLM-based judgments achieve strong agreement with human preferences across both correctness and consistency metrics.
>
>
> | Task Type  | Metric        | Correctness | Consistency |
> |------------|---------------|-------------|-------------|
> | **Overall**| Pairwise Acc  | 0.785       | 0.867       |
> |            | NDCG@7        | 0.924       | 0.962       |
> | **Global** | Pairwise Acc  | 0.826       | 0.887       |
> |            | NDCG@7        | 0.941       | 0.957       |
> | **Local**  | Pairwise Acc  | 0.766       | 0.874       |
> |            | NDCG@7        | 0.919       | 0.971       |
>
>
> > Response to W2: Presentation.
>
> We thank the reviewer for the suggestion. We will improve reproducibility by providing more explicit details on prompt generation and filtering heuristics in the final version.
>
> > Response to W3: Significance
>
> We thank the reviewer for the comment. While chart editing may appear specialized, it represents multimodal structured editing, requiring joint reasoning over visual layout, text, and data semantics. We believe this paradigm can generalize to domains such as documents, tables, and interfaces, with broader significance for multimodal model research.
>
> > Response to W4: Originality
>
> We thank the reviewer for the comment. Our primary contribution is indeed the benchmark and evaluation framework. We believe such benchmarks are essential for enabling and guiding future model research.
>
>
> > Response to Q1: To what extent does the benchmark rely on intermediate chart code when constructing ground-truth edits, and could this introduce bias into evaluation?
>
> We use intermediate chart code only for constructing controlled ground-truth edits, enabling precise specification of data, layout, and style changes.  Moreover, the generation pipeline spans diverse charts and tasks, reducing potential synthetic bias. In addition, we performed a strict manual filtering. We provide **stronger human validation as discussed in Response to W1**, which indicates that our evaluation is unlikely to introduce bias.
>
> > Response to Q2: How robust are the subjective evaluations performed by LLM judges compared to human annotators?
>
> To verify the robustness of LLM judges against low-level visual perturbations, we conducted a sensitivity analysis by applying slight perturbations to Nano Banana’s edited images, including Gaussian noise (std=0.3), image shifting (within 10px), and slight enlargement (1.02$\times$).
>
> | Condition | Cor. | Con. |
> | :--- | :---: | :---: |
> | Origin | 4.18 | 4.65 |
> | w/ Gaussian Noise | 4.24 | 4.70 |
> | w/ Shift | 4.19 | 4.64 |
> | w/ Enlarge | 4.20 | 4.65 |
>
> The results show that the **subjective metrics remained highly stable** despite these perturbations. This demonstrates that the LLM providing reliable and robust evaluations consistent with human-like judgment. Moreover, our results in **Response to W1** support this conclusion.
>
> > Response to Q3: Do failures on global editing primarily arise from visual reasoning limitations or from text rendering limitations in current models?
>
> We analyze this by manually sampling 100 failure cases that contain **at least one of the two error types** (visual reasoning or text rendering). The statistics are summarized below.
>
> | Model         | visual reasoning | text rendering |
> |---------------|------------------|----------------|
> | GPT-Image-1.5 | 100              | 3              |
> | Nano Banana   | 100              | 3              |
> | Qwen-Image    | 100              | 58             |
> | BAGEL         | 100              | 49             |
>
> Across all models, **visual reasoning errors consistently appear in all selected cases**, indicating that global editing failures are primarily driven by limitations in visual reasoning. In contrast, **text rendering errors are less frequent in proprietary models** (e.g., GPT-Image-1.5, Nano Banana, 3/100) but significantly more common in open-source models (e.g., Qwen-Image, BAGEL, 49–58/100).

---

> > ### Author Rebuttal · Reviewer_GBBG · 2026-04-04
> >
> > Thank you for the explanation provided by the authors. My concerns have been mainly addressed, and thus I will keep my positive score.

---

> > > ### Author Response · Authors · 2026-04-04
> > >
> > > We sincerely thank the reviewer for the valuable comments and efforts. We are glad that the concerns have been fully resolved.

---

### Decision · Program_Chairs · 2026-04-30

**Decision:**

Accept (regular)

**Comment:**

After reading the authors’ rebuttal and discussing intensively, all reviewers came to the consensus of accepting this paper. The AC agrees with the reviewers that this paper proposes a new and well-designed chart editing benchmark, and it did make some valuable contributions to the community.